# Quantitative theory for the diffusive dynamics of liquid condensates

Lars Hubatsch[1,2,3], Louise M Jawerth[1,2], Celina Love[2], Jonathan Bauermann[1], TY Dora Tang[2], Stefano Bo[1], Anthony A Hyman[2,3], Christoph A Weber[1,2,3]*

[1]Max Planck Institute for the Physics of Complex Systems, Dresden, Germany; [2]Max Planck Institute of Molecular Cell Biology and Genetics, Dresden, Germany; [3]Center for Systems Biology Dresden, Dresden, Germany

**Abstract** Key processes of biological condensates are diffusion and material exchange with their environment. Experimentally, diffusive dynamics are typically probed via fluorescent labels. However, to date, a physics-based, quantitative framework for the dynamics of labeled condensate components is lacking. Here, we derive the corresponding dynamic equations, building on the physics of phase separation, and quantitatively validate the related framework via experiments. We show that by using our framework, we can precisely determine diffusion coefficients inside liquid condensates via a spatio-temporal analysis of fluorescence recovery after photobleaching (FRAP) experiments. We showcase the accuracy and precision of our approach by considering space- and time-resolved data of protein condensates and two different polyelectrolyte-coacervate systems. Interestingly, our theory can also be used to determine a relationship between the diffusion coefficient in the dilute phase and the partition coefficient, without relying on fluorescence measurements in the dilute phase. This enables us to investigate the effect of salt addition on partitioning and bypasses recently described quenching artifacts in the dense phase. Our approach opens new avenues for theoretically describing molecule dynamics in condensates, measuring concentrations based on the dynamics of fluorescence intensities, and quantifying rates of biochemical reactions in liquid condensates.

*For correspondence:
christoph.weber@physik.uni-augsburg.de

Competing interests: The authors declare that no competing interests exist.

## Introduction

Liquid phase separation has emerged as an organizing principle in biology and is thought to underlie the formation of various membrane-less cellular organelles (*Banani et al., 2017*). Hallmark properties of such organelles are their rapid formation and dissolution, their fusion, and their wetting to membranes (*Hyman et al., 2014*). Moreover, phase-separated organelles exchange material with their environment leading to dynamic sequestration of molecules, which affects biochemical processes by spatial redistribution of reactants (*Moon et al., 2019*; *Lyon et al., 2021*; *Saha et al., 2016*; *Guillén-Boixet et al., 2020*; *Sanders et al., 2020*; *Yang et al., 2020*). Probing the dynamics of condensate components is thus imperative for a quantitative understanding of how they affect the cellular biochemistry (*Mir et al., 2019*).

To probe the dynamics of condensates, biomolecules are typically labeled with fluorescent tags. In general, in systems with tagged molecules, various methods exist to characterize molecular properties such as binding rates and diffusion coefficients, including fluorescence correlation spectroscopy (FCS) (*Ries and Schwille, 2012*; *Rigler and Elson, 2012*), single-particle tracking (SPT) (*Tinevez et al., 2017*; *Saxton and Jacobson, 1997*), and fluorescence recovery after photobleaching (FRAP) (*Diaspro, 2010*; *Stasevich et al., 2010*). However, interpretation of the experimental data acquired from such methods requires a rigorous derivation accounting for the underlying physico-chemcial processes. This derivation has been achieved for some biological systems and processes, but is lacking for condensates formed by liquid phase separation. Processes that are well-understood

include membrane-cytoplasmic exchange and transport (*Sprague et al., 2004*; *Robin et al., 2014*; *Goehring et al., 2010*) as well as chemical reactions (*Elson, 2001*) or filament turnover (*McCall et al., 2019*). For liquid condensates, various phenomenological fit functions have been proposed in the literature (e.g. *Patel et al., 2015*; *Banerjee et al., 2017*; *Hubstenberger et al., 2013*, for a broader summary see *Taylor et al., 2019*). However, it was recently shown that these fits lead to wildly differing estimates of the diffusion constant inside, $D_{\text{in}}$ (*Taylor et al., 2019*). Taylor et al. showed that these discrepancies were attributed to unrealistic assumptions, for example infinitely large droplets or infinitely fast diffusion outside the bleach area.

Here, we first introduce a quantitative FRAP method to extract the diffusion coefficient inside, $D_{\text{in}}$, purely based on fluorescence measurements inside droplets, without resorting to unrealistic assumptions or requiring knowledge about the partition coefficient, $P$ or diffusion outside, $D_{\text{out}}$. Using irreversible thermodynamics, we then derive the theory that connects dynamics inside and outside of the droplet via transport across a finite interface. We use the corresponding dynamic equation to derive a relationship between $P$ and $D_{\text{out}}$, which we use to investigate effects of salt addition on $P$. We show that this dynamic equation agrees with our experimentally observed dynamics. By numerically solving the underlying equations, we show that in theory all relevant parameters of the system, $P$, $D_{\text{in}}$, and $D_{\text{out}}$, can be extracted purely based on knowledge of the dynamics inside the droplet. We find that our measurements are agnostic to breaking radial symmetry, for example by introducing a coverslip or neighboring droplets. Our approach does not suffer from typical limitations of fluorescence-based concentration measurements, such as low fluorescence in the dilute phase and fluorophore quenching in the rich phase. We anticipate that this new understanding will open the door to characterizing dynamical properties such as chemical rates and rheological parameters in multi-component, phase-separated systems.

## Results

### Determining the diffusion constant inside liquid condensates

First, we discuss a quantitative method to extract diffusion coefficients of biomolecules in a condensate. After photobleaching, bleached molecules diffuse out and unbleached molecules diffuse into the condensate, until the unbleached components reach the spatially homogeneous level prior to bleaching (*Figure 1a*, left and middle). Inside a spherical condensate of radius $R$, the concentration of unbleached components, $c_u(r,t)$, follows a diffusion equation (for derivation, see subsequent section),

$$\partial_t c_u(r,t) = -\nabla \cdot \boldsymbol{j}_u, \tag{1a}$$

$$\boldsymbol{j}_u = -D_{\text{in}} \nabla c_u, \tag{1b}$$

$$c_u(r = R_-, t), \quad r = R, \tag{1c}$$

where $c_u(r = R_-, t)$ is the time-dependent concentration directly inside the interface at $r = R_-$. Here, $r$ denotes the radial distance to the center of the condensate. The flux $\boldsymbol{j}_u$ is given by Fick's law (*Equation (1b)*). It vanishes at the condensate center. Moreover, we have $\nabla = \boldsymbol{e}_r \partial_r$, with $\boldsymbol{e}_r$ denoting the radial unit vector. During FRAP, the concentration at the interface, $c_u(r = R_-, t)$, changes with time (*Figure 1a*, middle) and is determined by the physical properties of the condensate environment. This environment is characterized by the diffusion constant and the concentration of unbleached components outside, the distribution of neighboring condensates as well as system boundaries like the coverslip.

To initially bypass this complex dependence on the condensate environment, we propose to extract the fluorescence concentration of unbleached molecules $c_u(r = R_-, t)$ directly inside of the spherical phase boundary between the condensate and the dilute phase from experimental data after photobleaching (*Figure 1a*). To achieve spherical symmetry of the recovery inside the droplet, the entire spherical droplet should be bleached. Using the experimentally determined dynamic boundary condition $c_u(r = R_-, t)$, we can accurately determine the diffusion constant inside a condensate, $D_{\text{in}}$ (*Figure 1f,g*), provided $R^2/D_{\text{in}} \lesssim \tau_{\text{bound}}$. Here, $\tau_{\text{bound}}$ is the time scale of recovery at the

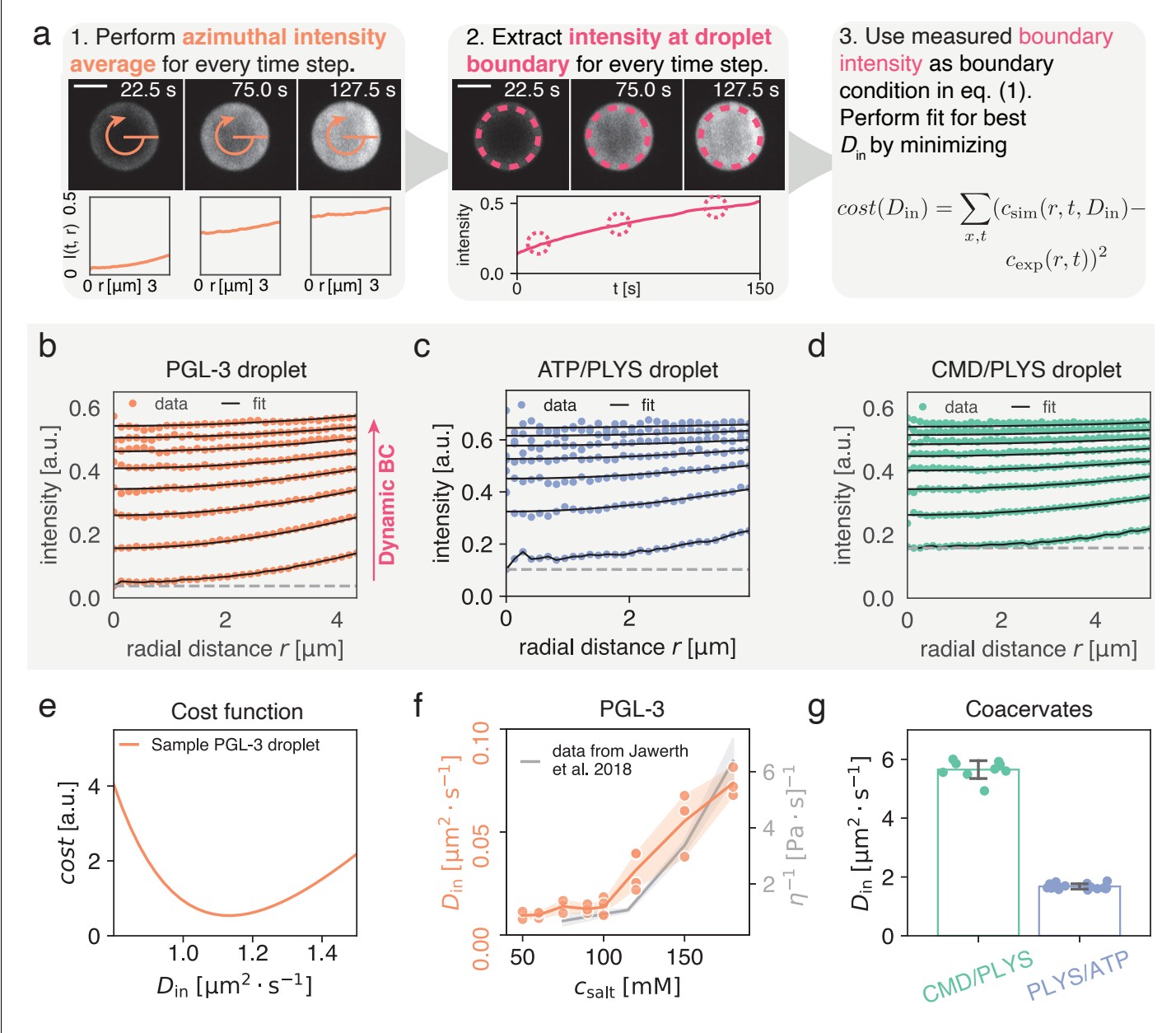

**Figure 1.** Quantitatively measuring $D_{in}$ by extracting intensity at condensate interface. (**a**) (1) Spatial time course of FRAP recovery after full bleach for PLYS/ATP coacervate droplet. Azimuthal average (orange) is highlighted. (2) To obtain the boundary condition $c(r = R_-, t)$, fluorescence intensity is extracted approximately $1.4\,\mu m$ away from the interface (see methods). Data closer to the droplet boundary can be subject to optical artefacts giving rise to an artificially broad interface. Lower panel shows azimuthal averages at different time points. (3) Cost function for fitting. For each fitting step, a numerical solution to *Equation (1a)* is calculated with a trial $D_{in}$ and the experimentally measured boundary condition from step (2). Subsequently simulation and experiment are compared according to the cost function, before choosing the next trial $D_{in}$. Scale bar, $5\,\mu m$. (**b**) Fluorescence recovery inside a PGL-3 condensate, along the radial direction (azimuthal averages, see (**a**)). $D_{in}$ is extracted by global fitting of *Equation (1b)* to the experimental profiles, using the experimentally extracted initial and boundary conditions (see panel (**a**) (1) and (2)). The gray line indicates an offset that comes about due to incomplete bleaching, and a small but visible, fast and uniform recovery with unknown cause (see Materials and methods). (**c**) Same as (**b**) but for ATP/PLYS coacervate droplet. (**d**) Same as (**b**) but for a CMD/PLYS coacervate droplet. (**e**) Cost function (see panel (**a**)) used to extract $D_{in}$ for a PGL-3 droplet. Due to the smoothness of the cost function, the minimum can be determined with high accuracy. (**f**) Comparison between $D_{in}$ and viscosity $\eta$ for PGL-3 condensates at different salt concentrations. Viscosity data taken from *Jawerth et al., 2018* for untagged PGL-3. Note, GFP-tagged PGL-3 has a higher viscosity than untagged PGL-3, which means the hydrodynamic radius of PGL-3:GFP cannot be directly computed from this panel (*Jawerth et al., 2020*). Replicates: salt concentration [mM]/#: 50/2, 60/3, 75/3, 90/3, 100/6, 120/4, 150/3, 180/3. (**g**) Diffusion coefficients for coacervate systems. Each data point represents $D_{in}$ for a single droplet time course. Note the low spread of the measured values. Replicates: CMD/PLYS: 9, PLYS/ATP:16.

boundary (see *Figure 1a*, middle), which in general features a complex dependence on bleach geometry in the dilute phase, the coverslip, and neighboring droplets. Following this idea, we fit the solutions of *Equation (1a)* to spatio-temporal experimental data, with $D_{in}$ as the only fit parameter. We find very good agreement between experimentally measured and fitted concentration profiles (*Figure 1b–d* and supp. *Videos 1*, *2* and *3*). Specifically, we consider condensates composed of PGL-3, a main protein component of P granules in the *C. elegans* embryo (*Brangwynne et al., 2009*; *Griffin, 2015*), as well as two synthetic polyelectrolye-complex coacervate systems, Polyly-sine/ATP (PLYS/ATP) and Carboxymethyldextran/Polylysine (CMD/PLYS).

We first compared $D_{in}$ of PGL-3 for different salt concentrations between $50\,mM$ and $180\,mM$ (see Methods). We find that $D_{in}$ varies by roughly one order of magnitude, between $0.009\,\mu m^2 s^{-1}$ and $0.070\,\mu m^2 s^{-1}$. Our trend is in agreement with reported measurements of the viscosity $\eta$, determined by active micro-rheology (*Jawerth et al., 2018*) for untagged PGL-3 (*Figure 1f*). Using viscosity data for GFP-tagged PGL-3 (*Jawerth et al., 2020*), we use the Stokes-Sutherland-Einstein relationship $D_{in} = k_B T/(6\pi a\eta)$ to estimate the hydrodynamic radius of PGL-3:GFP, $a = 1.5\,nm$ (*Einstein, 1905*; *Sutherland, 1905*; *von Smoluchowski, 1906*). This estimate is consistent with the value reported in *Liarzi and Epel, 2005*. Across all salt concentrations, the average coefficient of variation per condition is found to be $c_v = 0.22$. Due to the smooth cost function (*Figure 1e*) this is unlikely to be a stochastic artefact. It rather seems to reflect variation within the experimental assay.

For the coacervate systems, we find for the diffusion coefficients inside $D_{in} = 1.68 \pm 0.09\,\mu m^2 s^{-1}$ for PLYS/ATP coacervates and $D_{in} = 5.65 \pm 0.32\,\mu m^2 s^{-1}$ for CMD/PLYS coacervates; see (*Figure 1g*). The coefficient of variation of these measurements is low enough such that even a single measurement provides a good estimate of $D_{in}$. We find $c_v = 0.05$ and $c_v = 0.06$ for PLYS/ATP and CMD/PLYS coacervates respectively. Interestingly, $D_{in}$ for the coacervate droplets is about ten times smaller than the diffusion constant of the dilute polyelectrolytes, $D_{out}$ (*Arrio-Dupont et al., 1996*; *Morga et al., 2019*).

## Theory for the dynamics of labeled molecules in phase-separated systems

To understand the physical origin of the time-dependent concentration of unbleached components at the condensate interface and the phenomenological *Equations (1a, b, c)* for the dynamics inside a condensate, we need a theory that encompasses diffusion inside, outside and across phase boundaries. Here, we derive such a theory for a system that can be described by a binary, incompressible mixture prior to photobleaching. This binary mixture is composed of condensate material and solvent. The condensate material has a concentration profile, $c_{tot}(\boldsymbol{x},t) = \phi_{tot}(\boldsymbol{x},t)/\nu$, which can be expressed in terms of a volume fraction profile $\phi_{tot}(\boldsymbol{x},t)$ by dividing through the molecular volume of the condensate material, $\nu$. Due to incompressibility, the solvent volume fraction in such a binary mixture is given by $(1 - \phi_{tot})$. The system prior to photobleaching is assumed to be either at equilibrium, that is, a single droplet and $\partial_t\phi_{tot}(\boldsymbol{x},t) = 0$, or close to equilibrium, that is, a system composed of many droplets undergoing slow Ostwald ripening and fusion and $\partial_t\phi_{tot}(\boldsymbol{x},t) \simeq 0$. Thus, the (quasi-)stationary profile $\phi_{tot}(\boldsymbol{x})$ prescribes a physical constraint for FRAP dynamics.

After photobleaching, the system becomes a *ternary* incompressible mixture composed of bleached (*b*) and unbleached (*u*) components, as well as solvent (*Figure 2a*). Introducing the volume fraction of the bleached and unbleached components, $\phi_b$ and $\phi_u$, the physical constraint for the FRAP dynamics reads

$$\phi_{tot}(\boldsymbol{x},t) = \phi_u(\boldsymbol{x},t) + \phi_b(\boldsymbol{x},t), \qquad (2)$$

where the profiles depend on space and time. The above constraint reflects particle number conservation of bleached and unbleached molecules and becomes a local constraint due to incompressibility. In our work, we focus on FRAP

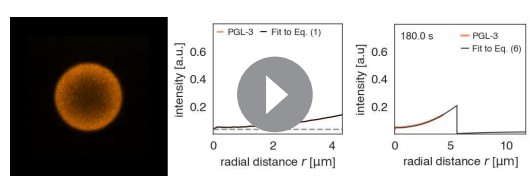

**Video 1.** FRAP dynamics in a PGL-3 droplet Left: Representative in vitro droplet after full bleach. Time course starts after a small time lag due to a fast uniform recovery (see methods). Middle: Diffusion *Equation (1)* fit to azimuthally averaged droplet intensity with a global fit parameter $D_{in}$. Right: Full model *Equation (6)* fit to azimuthally averaged droplet intensity.

https://elifesciences.org/articles/68620#video1

dynamics at thermodynamic equilibrium, where the total volume fraction $\phi_{\text{tot}}(\boldsymbol{x})$ depends on space only. Please note that that the derivation below can be generalized to non-equilibrium situations where $\phi_{\text{tot}}$ exhibits a flux (**Bo et al., 2021**).

Immediately after photobleaching, unbleached molecules diffuse into the condensate leading to FRAP dynamics of unbleached molecules inside (**Figure 2b,c**). At long times, the concentration profile of unbleached molecules approaches the profile prior to photobleaching, $\phi_{\text{tot}}(\boldsymbol{x})$. The dynamics of both concentration profiles, bleached and unbleached molecules, $c_i = \phi_i/\nu_i$ ($i = b, u$), with $\nu_i$ denoting the molecular volumes, is described by the following conservation laws ($j = u, b$),

$$\partial_t c_i = -\nabla \cdot \boldsymbol{j}_i,$$ (3a)

$$\boldsymbol{j}_i = -\Gamma_i \nabla \mu_i - \bar{\Gamma}_{ij} \nabla \mu_j,$$ (3b)

Here, $\Gamma_i$ are the Onsager transport coefficients, often referred to as mobilities, and $\bar{\Gamma}_{ij}$ are the Onsager cross coupling coefficients obeying $\bar{\Gamma}_{ij} = \bar{\Gamma}_{ji} =: \bar{\Gamma}$. In general, both mobility coefficients depend on the volume fractions. According to irreversible thermodynamics, the flux $\boldsymbol{j}_i$ is driven by gradients in exchange chemical potentials $\mu_i$ and $\mu_j$. In the following, we consider linear response for simplicity. The exchange chemical potentials, $\mu_i = \delta F/\delta c_i$, are linked to the free energy, $F = \int d^3x f$, where $f$ denotes the free energy density. Expressing concentrations in terms of the volume fractions, $\phi_i = c_i \nu_i$, we describe our incompressible ternary mixture after photobleaching by a Flory-Huggins free energy density (**Flory, 1942**; **Huggins, 1942**; **Krüger et al., 2018**):

$$f = \frac{k_B T}{\nu_{\text{sol}}} \left[ \frac{\phi_u}{n_u} \ln \phi_u + \frac{\phi_b}{n_b} \ln \phi_b + (1 - \phi_u - \phi_b) \ln(1 - \phi_u - \phi_b) + \chi_{us} \phi_u (1 - \phi_u - \phi_b) \right.$$
$$\left. + \chi_{bs} \phi_b (1 - \phi_u - \phi_b) \quad + \chi_{ub} \phi_u \phi_b + \frac{\kappa_u}{2} (\nabla \phi_u)^2 + \frac{\kappa_b}{2} (\nabla \phi_b)^2 + \frac{\kappa_{ub}}{2} \nabla \phi_u \cdot \nabla \phi_b \right],$$ (4)

where we write the molecular volumes of bleached and unbleached components in $n_i$ multiples of the solvent molecular volume $\nu_{\text{sol}}$, that is, $\nu_i = n_i \nu_{\text{sol}}$. Moreover, $\chi_{ij}$ denote dimensionless Flory-Huggins parameters characterizing the interactions between different components $i$ and $j$, where subscript $s$ indicates the solvent. The parameters $\kappa_i$ and $\kappa_{ub}$ characterize the free energy penalties for spatial heterogeneities and are linked to the surface tensions.

If photobleaching does not affect the molecular interactions or molecular volumes, the free energy density above can be simplified significantly (Appendix of **Krüger et al., 2018**). In this case, the interactions between unbleached and solvent, and bleached and solvent components are equal, $\chi_{us} = \chi_{bs} =: \chi$, and cross interactions vanish, $\chi_{ub} = 0$. Moreover, molecular volumes of bleached and unbleached components are equal, $n_u = n_b =: n$, and the parameters characterizing free energy penalties for spatial heterogeneities obey $\kappa_u = \kappa_b =: \kappa$ and $\kappa_{ub} = 2\kappa$. Thus, the simplified free energy reads

$$f = \frac{k_B T}{\nu_{\text{sol}}} \left[ \frac{\phi_u}{n} \ln \phi_u + \frac{\phi_b}{n} \ln \phi_b + (1 - \phi_u - \phi_b) \ln(1 - \phi_u - \phi_b) + \chi(1 - \phi_u - \phi_b)(\phi_u + \phi_b) + \frac{\kappa}{2} (\nabla(\phi_u + \phi_b))^2 \right]$$ (5)

To ensure a constant diffusion coefficient in the dilute limits of the bleached and unbleached components, we employ the scaling ansatz for a ternary mixture, $\Gamma_i = \Gamma_0 \phi_i \left[ (1 - \phi_{\text{tot}}) + (\bar{\Gamma}_0/\Gamma_0)\phi_j \right]$ and $\bar{\Gamma} = -\bar{\Gamma}_0 \phi_u \phi_b$. In general, both mobility functions, $\Gamma_0$ and $\bar{\Gamma}_0$, depend on the total volume fraction $\phi_{\text{tot}}$. For the limiting case where bleached and unbleached molecules are identical particles, we can choose $\Gamma_0 = \bar{\Gamma}_0$. Applying the equilibrium FRAP condition (2) and using **Equation (3a, b)**, we find that the concentration of unbleached components is governed by the following diffusion equation

$$\partial_t c_u = \nabla \cdot \left[ D(\phi_{\text{tot}}) \left( \nabla c_u - c_u \frac{\nabla \phi_{\text{tot}}}{\phi_{\text{tot}}} \right) \right],$$ (6)

with a $\phi_{\text{tot}}(\boldsymbol{x})$-dependent diffusivity, $D(\phi_{\text{tot}}) = k_B T \Gamma_0(\phi_{\text{tot}})$. As we show in **Bo et al., 2021** a similar approach can be used to investigate single-molecule dynamics across phase boundaries.

Similar to **Equation (1)**, the diffusion equation above is linear in $c_u$. However, the dynamics of unbleached components are affected by gradients in $\phi_{\text{tot}}(\boldsymbol{x})$ and components diffuse with different diffusion coefficients inside and outside the condensate, where in each phase $\nabla \phi_{\text{tot}} = 0$ (**Figure 2b**).

**Video 2.** FRAP dynamics in a CMD/PLYS coacervate. For description see *Video 1*.

https://elifesciences.org/articles/68620#video2

The position-dependence of $\phi_{tot}(\mathbf{x})$ is given by the equilibrium condition of a homogeneous chemical potential of the binary mixture prior to photobleaching, which implies $\partial_t \phi_{tot} = 0$. For a radially symmetric system with $r$ denoting the radial coordinate, $\phi_{tot}(r) = \phi_{out}^{eq} + (\phi_{in}^{eq} - \phi_{out}^{eq})H((r-R)/\ell)$, where $\phi_{in}^{eq}$ and $\phi_{out}^{eq}$ are the equilibrium volume fractions inside and outside, respectively, $R$ is the droplet radius, and $\ell$ denotes the width of the interface. Moreover, $H((r-R)/\ell)$ is a function that decreases from one to zero at $r = R$ on an interface width $\ell$. For phase separation close to the critical point and large droplet sizes, $H(x) = (1 + \tanh(x))/2$ (*Bray, 1994*; *Weber et al., 2019*). We numerically solve *Equation (6)* using a finite element method (*Anders et al., 2012*) in a finite domain of size $L$ which is much larger than the droplet radius $R$ and fit the solution to experimental data.

In summary, our model has seven parameters. Four of these, namely, the equilibrium volume fractions $\phi_{in}^{eq}$ and $\phi_{out}^{eq}$, the interface width $\ell$, and the droplet radius $R$, characterize the equilibrium profile prior to bleaching $\phi_{tot}(\mathbf{x})$. The remaining parameters are the system size $L$ and the diffusion coefficients inside and outside, which are given as

$$D_{in} = k_B T \Gamma_0(\phi_{in}^{eq}), \tag{7a}$$

$$D_{out} = k_B T \Gamma_0(\phi_{out}^{eq}). \tag{7b}$$

For the case of a single, spherical droplet with an infinitely thin interface (*Weber et al., 2019*), we can derive an effective droplet model for the unbleached component from *Equation (6)*, where the dynamics of unbleached components inside and outside are given by diffusion equations that are coupled by boundary conditions (see Appendix 1 for the derivation):

$$\partial_t c_u(r,t) = D_{in} \nabla^2 c_u, \quad \text{for } r < R, \tag{8a}$$

$$\partial_t c_u(r,t) = D_{out} \nabla^2 c_u, \quad \text{for } r > R, \tag{8b}$$

$$-D_{in} \mathbf{e}_r \cdot \nabla c|_{R_-} = -D_{out} \mathbf{e}_r \cdot \nabla c|_{R_+}, \tag{8c}$$

$$c_u(r = R_-, t) = P c_u(r = R_+, t). \tag{8d}$$

Here, $R_-$ and $R_+$ denote the radial position directly inside and outside the droplet interface, respectively. *Equation (8c)* describes an equality of the fluxes directly inside and outside of the interface, respectively, and thereby expresses particle number conservation at the interface $r = R$. *Equation (8d)* describes a jump in concentration of unbleached components, which is determined by the thermodynamic partition coefficient

$$P = \frac{\phi_{in}^{eq}}{\phi_{out}^{eq}}. \tag{9}$$

**Video 3.** FRAP dynamics in a PLYS/ATP coacervate. For description see *Video 1*.

https://elifesciences.org/articles/68620#video3

Moreover, the flux vanishes at the origin $r = 0$, $\mathbf{e}_r \cdot \nabla c|_{r=0} = 0$, and at the system boundary $r = L$, $\mathbf{e}_r \cdot \nabla c|_{r=L} = 0$.

Above we provide a thermodynamic derivation of *Equations (8)* in the limit of thin interfaces. These equations were already proposed as a model for FRAP dynamics of protein condensates (*Taylor et al., 2019*). Interestingly, while we

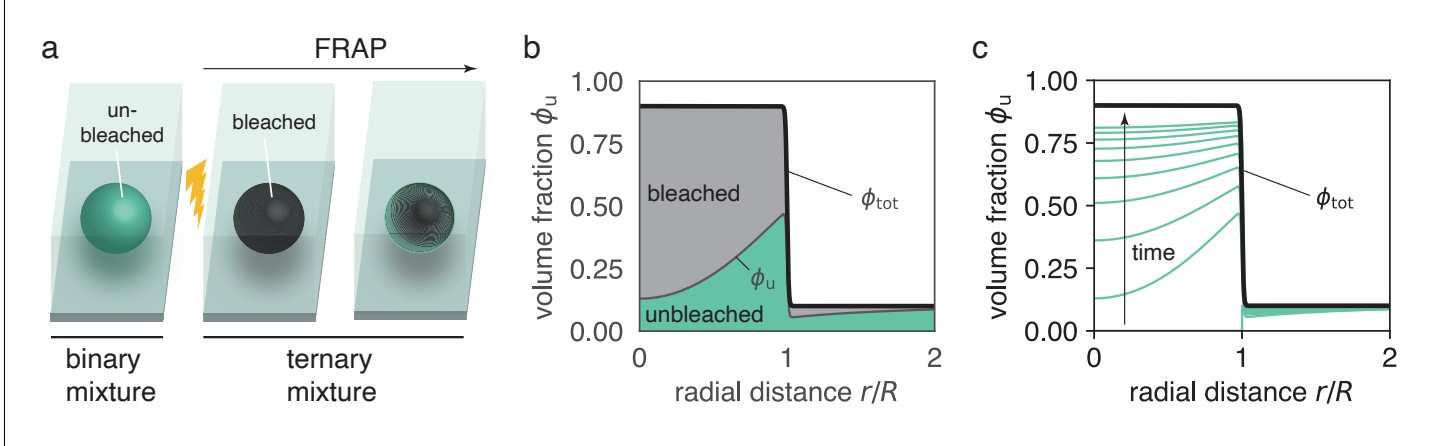

**Figure 2.** Ternary mixture accounts for the dynamics of bleached and unbleached molecules. (a) Before bleaching, a droplet that is composed of fluorescently labeled molecules can be described by a binary mixture, namely unbleached molecules and solvent. After bleaching, the system is composed of three components, bleached molecules, unbleached molecules and solvent. If the system was at equilibrium prior to bleaching, the sum of bleached and unbleached molecules forms a stationary, non-uniform profile $\phi_{tot}(r)$ (see panel b). (b) Snapshot of model dynamics at $t = 0.22R^2/D_{in}$. Initial conditions are $\phi_u(r, t = 0) = \phi_{out} \cdot \Theta(r - R)$, corresponding to a fully bleached droplet. Note that at any time we have $\phi_{tot} = \phi_u + \phi_b$. (c) Time course of spatial recovery. For long times, when nearly all bleached material has been exchanged, $\phi_u$ approaches $\phi_{tot}$. Panels (c,d) use radial symmetry for illustration purposes, however, the theory is general (see *Figure 3*).

obtained *Equations (8)* for the dynamics of a phase-separated protein component undergoing fluorescence recovery, similar equations were also used to investigate diffusion of a protein that was added to an already existing two-phase system at thermodynamic equilibrium (*Münchow et al., 2008*).

In *Equations (8)* $D_{in}$, $D_{out}$, and $P$ are considered to be independent parameters. Strictly speaking, due to phase separation, the diffusion coefficients $D_{in}$ and $D_{out}$ are not independent which is evident in *Equations (7)*. For example, in the absence of phase separation or at the critical point, $\phi_{in}^{eq} = \phi_{out}^{eq}$ (i.e. $P = 1$), the diffusion coefficients inside and outside must be equal, $D_{in} = D_{out}$. For a given condensate with fixed $\phi_{in}^{eq}$ and $D_{in}$, there is a relationship between the diffusivity outside $D_{out}$ and the partition coefficient $P$, which can be expressed using *Equations (7)* as

$$D_{out}(P) = D_{in} \frac{\Gamma_0(\phi_{in}^{eq})}{\Gamma_0(\phi_{in}^{eq}/P)} . \tag{10}$$

However, except for the limit $P \to 1$, *Equation (10)* does not impose further constraints for the determination of the parameters since the mobility function $\Gamma_0(\phi_{tot})$ is unknown. For large $P$, the missing knowledge of the mobility function renders $D_{out}$, $D_{in}$ and $P$ as effectively independent parameters. This provides a theoretical justification for the assumption made by *Taylor et al., 2019*.

In the following, we use our theory (*Equation (6)*) to investigate the impact of the condensate environment on the FRAP dynamics. In particular, we consider how a passivated coverslip (no wetting of condensates) and nearby condensates affect the influx of unbleached molecules and thereby the recovery dynamics. Lastly, given the concentration at the droplet boundary, $c(r = R, t)$, we derive a relationship between $D_{out}$ and $P$. This can be used to investigate changes of the partition coefficient, for example when changing salt concentrations. Importantly, this method does not rely on absolute fluorescence intensities when measuring concentrations in the dense and dilute phases, which are prone to artefacts (*McCall et al., 2020*).

## Impact of non-wetting coverslip on FRAP dynamics

Here, we investigate the influence of the coverslip surface on the FRAP dynamics of non-wetting spherical droplets. Under typical in vitro conditions, condensates sediment due to gravity, leading to sessile droplets on a coverslip. In many experimental setups, coverslips are passivated, for example pegylated, in order to suppress wetting of condensates on the coverslip surface (*Alberti et al., 2018*). These experimental conditions lead to almost spherical droplets since capillary effects are

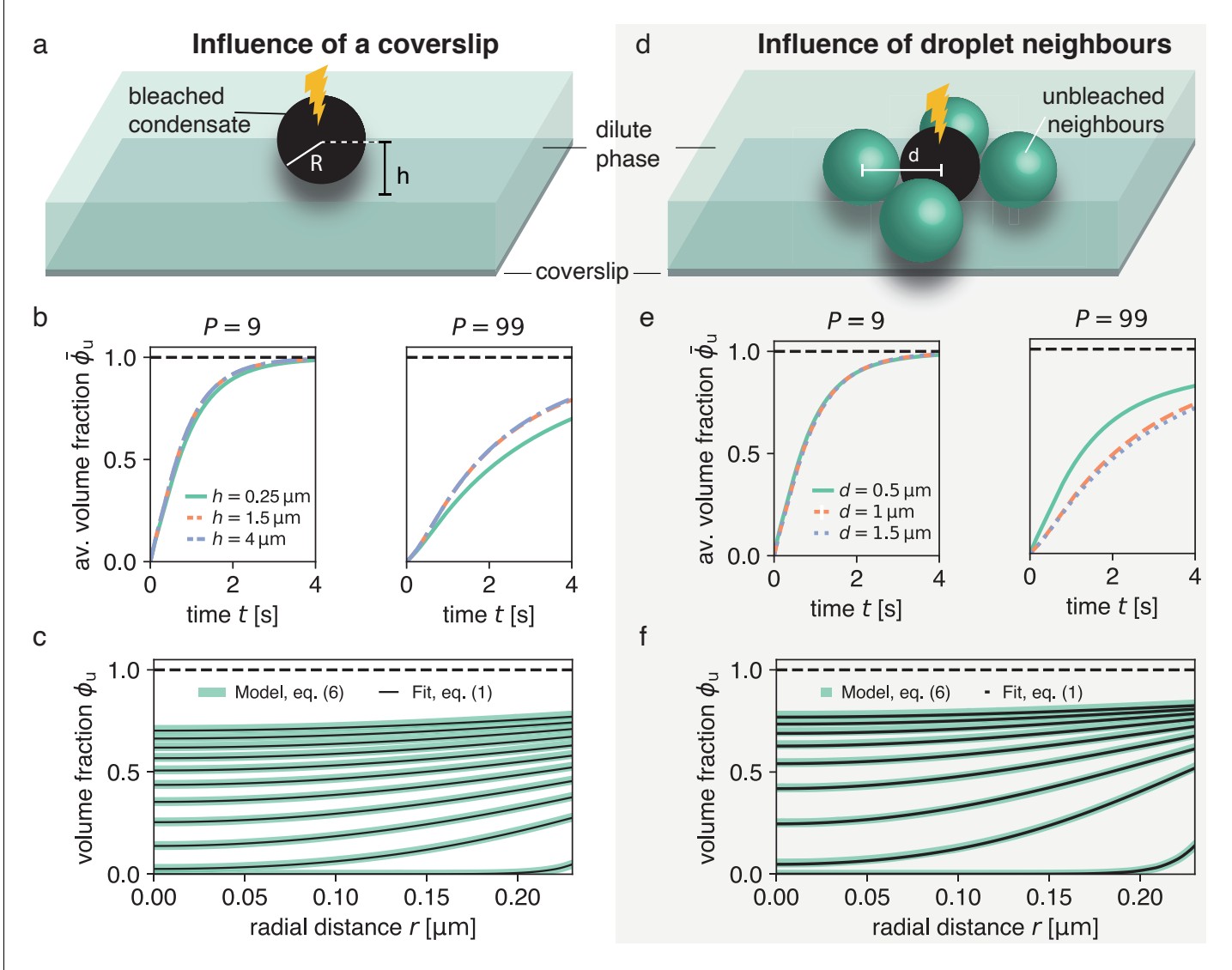

**Figure 3.** Impact of droplet environment on recovery dynamics. (a) Sketch of a typical experimental set-up with a droplet above a passivated coverslip, where the droplet center has a distance $h>R$ to the coverslip. (b) Recovery of average unbleached volume fraction $\bar{\phi}_u(t) = \int d^3r\, \phi_u(r,t) \cdot 3/(4\pi R^3)$ for different heights $h$ above the coverslip at different partition coefficients $P$. Results were obtained by solving **Equation (6)** using the finite element method and considering the geometries depicted in (a). For even larger $h$-values (e.g. no coverslip), results are approximately equal to the blue dashed line. (c) Using the method introduced in **Figure 1** on the scenario with the largest influence of the coverslip (droplet sessile on coverslip) $h = 0.25\,\mu m$ in (a) results in an excellent fit and can reliably extract the input $D_{in}$. (d) Sketch of neighboring droplets next to a bleached droplet. (e) Total recovery curves for finite element simulations of the geometry depicted in (d), for different distances between neighboring droplet centers, at different partition coefficients $P$. Note the strong dependence on the distance of neighboring droplets. For even larger $d$-values (e.g. no neighboring drops), results are approximately equal to the blue dashed line. (f) Same as (c) but for largest influence of neighboring droplets, that is $d = 0.5\,\mu m$, where there is no distance between bleached droplet and neighboring droplets.

typically negligible for micrometer-sized polymer-rich or protein-rich droplets (**Park et al., 2013**; **McCall et al., 2020**).

We numerically solved **Equation (6)** for a spherical condensate on top of a no-flux boundary that mimics the coverslip surface (**Figure 3a**). We find that the recovery of the average volume fraction inside the condensate can slow down compared to the case without a coverslip (**Figure 3b**). This slow-down vanishes if droplets have a distance to the coverslip surface larger than a few droplet diameters. Moreover, we find that the slow-down is more pronounced for larger partition coefficients $P$. This trend can be explained by the size of the region in the dilute phase from where most

unbleached molecules are recruited: if $P$ is small, most of the unbleached molecules come from the immediate surroundings of the condensate. Hence the influence of radial asymmetry is minimal and the recovery appears almost unchanged compared to the case without coverslip. However, for large $P$, the condensate recruits unbleached molecules from distances far away, limiting the recruitment to an effective half-space compared with the case without coverslip. Indeed, for very large $P$ recovery rates slow down maximally by a factor of two.

Interestingly, by extracting the boundary concentration in mid-plane, similar to the procedure in *Figure 1*, and spatially fitting the solutions of *Equation (1a)* to the ensuing recovery profiles, we can reliably recover the input diffusion coefficient $D_{in}$ (*Figure 3c*). The reason for this agreement is that by considering the intensity at the condensate interface, our method is independent of the time scale set by diffusion in the dilute phase.

## Impact of neighboring condensates on FRAP dynamics

In this section, we address the impact of neighboring condensates on the recovery dynamics. We solved *Equation (6)* for a system composed of a bleached condensate with four unbleached neighboring condensates of the same size (*Figure 3d*) and find that neighboring condensates can significantly speed up the recovery dynamics (*Figure 3e*). This speed-up is only pronounced for rather close condensates with inter-droplet distance on the order of condensate size. Moreover, similar to the impact of the coverslip, the effects of the recovery dynamics are stronger for larger partition coefficients $P$. For small $P$, most unbleached molecules are recruited from the dilute phase leading to almost no effect on the recovery also when condensates are very close to each other (*Figure 3e*, left). For large partition coefficients $P$, however, a certain fraction of molecules are recruited from the neighboring condensates causing a significant speed-up of the recovery (*Figure 3e*, right).

Again, despite this change in total recovery due to close-by neighboring droplets, we can reliably measure $D_{in}$ via our spatial fitting method (*Figure 3f*). In particular, by extracting the boundary intensity in mid-plane and spatially fitting the solutions of *Equation (1a)*, we find very good agreement with our input $D_{in}$. This agreement shows that our method is robust for typical experimental systems that deviate from an ideal, isolated condensate.

## Interfacial flux relates partition coefficient and outside diffusivity

We have shown that by using the time-dependent fluorescence at the interface of a spherical droplet we can accurately fit our dynamic *Equation (1)* to our experimental data and thus determine the diffusion constant inside the droplet, $D_{in}$ (*Figure 1a*). Our theory (see *Equation 6*) suggests that the fluorescence at the droplet interface is affected by the physical parameters characterizing the droplet environment such as the diffusion coefficient $D_{out}$ and the partition coefficient $P$. In particular, the flux through the droplet interface is enlarged for increasing $D_{out}$ or decreasing $P$ (see *Equation (8c)* after rescaling the concentration close to the interface). Thus, for a condensate with concentration $\phi_{in}^{eq}$ and diffusion coefficient $D_{in}$, a given flux between both phases through the interface implies a relationship between $D_{out}$ and $P$.

Here, we determined the relationship between $D_{out}$ and $P$ by fitting numerical solutions of *Equation (6)* to the recovery dynamics inside the droplet (*Figure 4b–d* and *Videos 1*, *2* and *3*). The diffusion coefficient inside, $D_{in}$, was independently determined for each experiment via our method introduced in *Figure 1a*. This leaves $P$ and $D_{out}$ as independent parameters, which is valid for large $P$ (see discussion after *Equation (10)*). We thus sampled $P$ (with $P \gg 1$) across three orders of magnitude and obtained the best-fitting $D_{out}$ for each $P$ (*Figure 4e,f*). Notably, all the combinations of $D_{out}$ and $P$ represent relatively good fits, and each experimental condition leads to a unique relationship $D_{out}(P)$. For large $P$, we find that the best $D_{out}$ scales linearly with $P$ (dashed lines in *Figure 4e, f*); for a discussion on the origin of this scaling, please refer to Appendix 2.

We would like to draw attention to the values of $P$, which are higher than expected (A.W. Fritsch and J.M. Iglesias-Artola, personal communication), for a realistic parameter range $50\ \mu\mathrm{m}^2\mathrm{s}^{-1} < D_{out} < 100\ \mu\mathrm{m}^2\mathrm{s}^{-1}$. One reason for this discrepancy is the assumption of spherical symmetry for the fitting routine leading to absolute values of $P$ that are overestimated up to twofold (see caption of *Figure 3* for a discussion). Additionally, our current experiments cannot exclude boundary effects, such as a recently hypothesised interfacial resistance (*Taylor et al., 2019*). *Equation (6)*

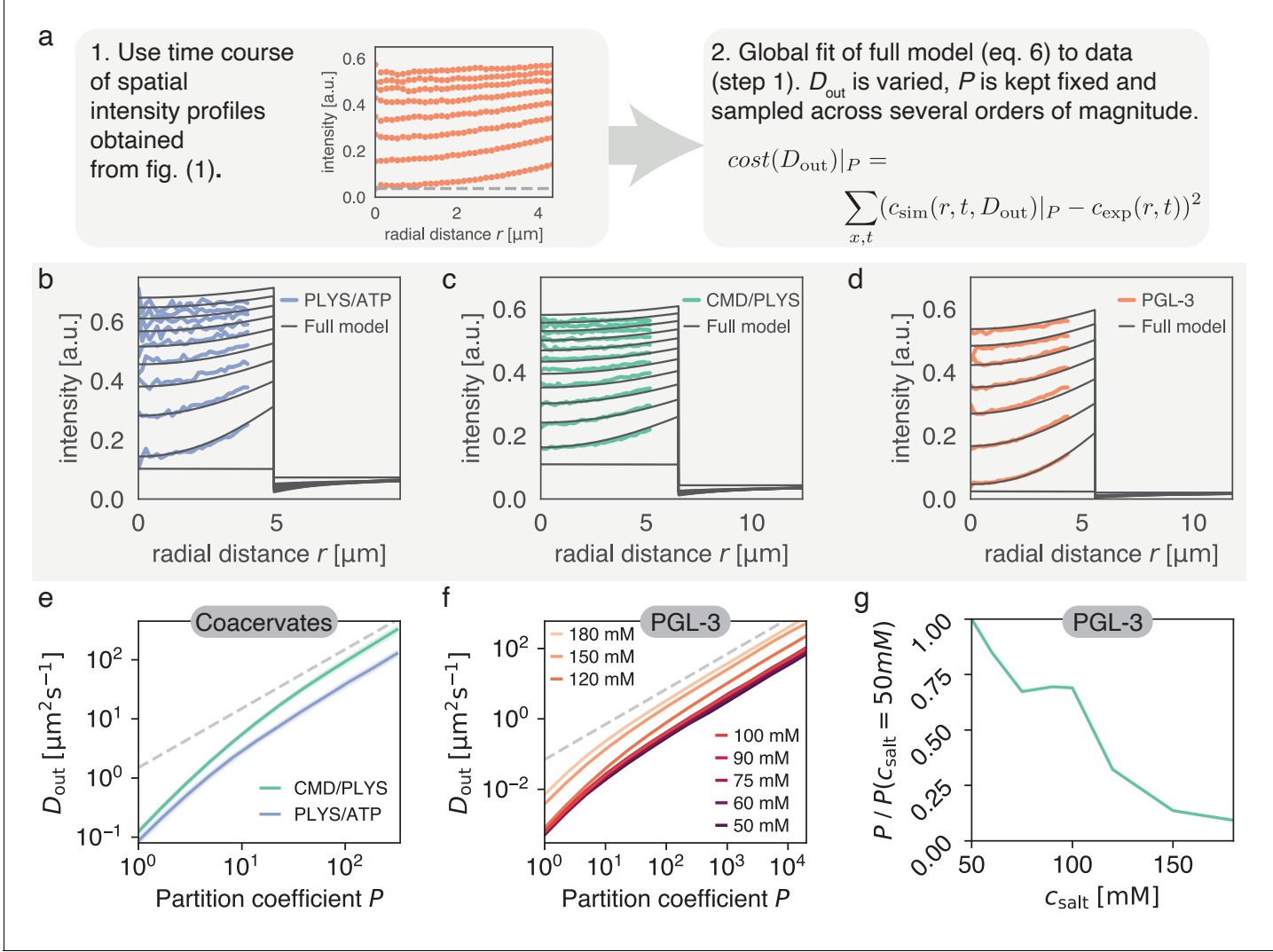

**Figure 4.** Varying partition coefficient $P$ and diffusivity outside $D_{out}$ simultaneously can lead to similar recovery kinetics. (a) Schematics of the fitting approach used to extract the relationship between $P$ and $D_{out}$. Although there is one specific pair ($P$, $D_{out}$) with the lowest cost (see *Figure 5*), there is a line in ($P$, $D_{out}$) space, which produces good fits for a range of values. (b) Spatial recovery of a single PLYS/ATP droplet (blue) with a fit to the full model (black). Note, data close to the droplet boundary cannot be fit, due to optical artefacts giving rise to an artificially broad interface (see Materials and methods and *Figure 1a*). (c) Same as (b) for a CMD/PLYS coacervate droplet. (d) Same as (b) for a PGL-3 droplet. (e) Given the partition coefficient $P$, $D_{out}$ is found by fitting the coacervate data to the model. Note the convergence to a power law, $D_{out} \propto P^n$ with $n = 1$ for large partition coefficients (gray dashed line; for a discussion see Appendix 2). Shaded area around curves: standard deviation. Replicates: same as in *Figure 1g*. (f) Same as (e) but for PGL-3 with different salt concentrations. Note the order from top to bottom from highest to lowest salt concentration. (g) Based on (f), the change of partition coefficient $P(c_{salt})$ can be estimated for a given $D_{out}$. Confidence intervals not shown for clarity. Similar to panel e. Replicates: same as in *Figure 1f*.

should therefore be interpreted as a minimal model that fits the available data with high accuracy and explains the boundary dynamics, but cannot rule out additional effects.

Thus, the relationship $D_{out}(P)$ has to be assessed critically. Since $P$ and $D_{out}$, in conjunction with $D_{in}$, set the time scale for recovery at the boundary, any effects that are ignored would change this relationship. Since the boundary time scale is well-described by $D_{out} \propto P$, this change would likely manifest itself as a constant prefactor in each curve in *Figure 4e,f*. We thus only interpret the ratio between curves in *Figure 4e,f*, instead of their absolute values.

Considering this ratio allows us to investigate how the addition of salt affects the partition coefficient of PGL-3, assuming that boundary effects and $D_{out}$ are independent of $c_{salt}$. *Figure 4f* indicates that for all salt concentrations we have $D_{out} \propto P$ for $D_{out} > 1 \, \mu m^2 s^{-1}$. For proteins and coacervate

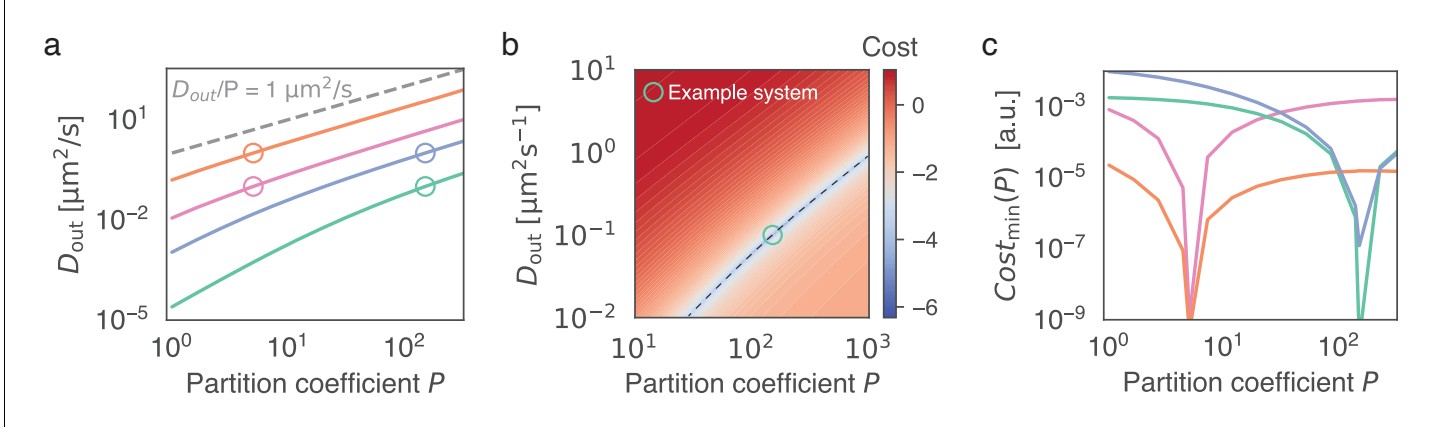

**Figure 5.** Given the recovery dynamics inside a condensate, the key parameters $D_{in}$, $D_{out}$ and $P$ can in theory be determined uniquely without measuring the outside dynamics, using *Equation (6)*. (a) Given the partition coefficient $P$, $D_{out}$ is found by fitting the model to synthetically generated example data. As example systems, indicated by open circles, we consider in silico data, obtained by solving *Equation (6)* with known parameters $P$ and $D_{out}$. To mimic the approach of initially determining $D_{in}$ (see *Figure 1*) we keep $D_{in} = 0.01\ \mu m^2 s^{-1}$ for all in silico datasets. For each example system (open circles), the best fitting $D_{out}$ is found given a range of $P$ (solid lines). Parameters used were as follows: pink circle: $D_{out} = 0.1\ \mu m^2 s^{-1}$, $P = 5$, orange circle: $D_{out} = 1\ \mu m^2 s^{-1}$, $P = 5$, green circle: $D_{out} = 0.1\ \mu m^2 s^{-1}$, $P = 150$, blue circle: $D_{out} = 1\ \mu m^2 s^{-1}$, $P = 150$. (b) Cost function (colorbar, log-scale) as a function of $D_{out}$ and $P$. We note that the global minimum coincides with the parameters used to generate the synthetic data (green circle). The valley in parameter space (dashed line) corresponds to the green line in (a). (c) Minimum of cost function for each $P$, corresponding to curves shown in (a). This minimum corresponds to the valley indicated by the dashed line in (b). Note the minimum at the input parameter set, which indicates uniqueness of the outside dynamics for given values of $D_{out}$ and $P$.

components we can safely assume $D_{out} > 10\ \mu m^2 s^{-1}$ in solution. Consequently, we chose $D_{out} = 10\ \mu m^2 s^{-1}$ and calculated how the partition coefficient $P$ changes with salt concentration $c_{salt}$ (*Figure 4g*). Specifically, for salt concentrations in the range from 50 mM to 180 mM, we find that the estimated partition coefficient $P$ of PGL-3 droplets decreases approximately 10-fold. This trend is probably a result of enhanced screening of charged groups for increasing salt concentration.

## Proposal how to determine partition coefficient and outside diffusivity

We next asked, whether we can obtain both parameters, $P$ and $D_{out}$, at the same time, without measuring fluorescence intensities outside the droplet. Although each combination ($D_{out}$, $P$) along the lines specified in *Figure 4e,f* leads to a reasonable fit, we will now show that there is a distinct combination that exhibits a global minimum of the cost function for each $D_{out}(P)$. Here, we assume no resistance at the interface as proposed in *Münchow et al., 2008*; *Hahn et al., 2011*; *Hahn and Hardt, 2011*. Providing experimental evidence of this global minimum can be hampered by effects due to the droplet environment, such as neighboring droplets or the coverslip surface (*Figure 3a,d*), inhomogeneous bleaching in the dilute phase, imaging artefacts at the phase boundary and effects at the droplet boundary not accounted for in *Equation (6)*. In particular, in our experimental studies, interdroplet distances are sometimes on the order of the droplet size and diffusive exchange is affected by the coverslip. Thus, we decided to use our model to create in silico data and provide evidence for the existence of a distinct combination of $D_{out}$ and $P$ for a fixed $D_{in}$. Fixing $D_{in}$ mimics the approach of initially determining $D_{in}$, as outlined in *Figure 1*. To determine the relationship $D_{out}(P)$, we proceed as described for *Figure 4*. *Figure 5a* depicts the $D_{out}(P)$ relationships corresponding to four parameter combinations in a range relevant for protein condensates and coacervate droplets (*Drobot et al., 2018*; *Riback et al., 2020*). In particular, we choose two outside diffusivities $D_{out}$ of $0.1\ \mu m^2 s^{-1}$ and $1\ \mu m^2 s^{-1}$ and two partition coefficients, $P = 5$ and $P = 150$. We find indeed that each cost function exhibits a unique minimum for each of the considered parameter combinations (*Figure 5b,c*). These findings indicate that all three parameters, $D_{in}$, $D_{out}$ and $P$, can in theory be determined by a single FRAP time course of the droplet intensity inside. Thus, in principle, there is no need to measure kinetic properties of the dilute phase to fully characterise the system in terms of its parameters. This possibility represents a new approach to characterise the partition coefficient $P$,

which is particularly important in light of recent data showing that measurements based on fluorescence intensity can lead to drastic underestimation of $P$ (*McCall et al., 2020*).

## Discussion

The dynamic redistribution of fluorescent molecules has been used to characterize liquid phase separation in biology via a variety of techniques, including SPT, FRAP, and FCS (*Elbaum-Garfinkle et al., 2015*; *Taylor et al., 2019*; *Moon et al., 2019*). Here, we have derived a theory that describes the diffusive motion of labeled molecules based on the physics of phase separation. It can be applied to many state-of-the art fluorescent methods such as FCS, SPT, and FRAP and can thus help extend traditional techniques to the realms of phase separation (*Ries and Schwille, 2012*). Importantly, this theory enables us to avoid commonly applied approximations such as the frequently used single-exponential recovery (*Brangwynne et al., 2009*; *Frottin et al., 2019*; *Kaur et al., 2019*; *Fisher and Elbaum-Garfinkle, 2020*; *Kistler et al., 2018*).

Our theory, essentially represented by *Equation (6)*, governs the dynamics of labeled molecules through interfaces of condensates. As we show in our work, it can be applied to spherical condensates. In addition, our theory could also be used for non-spherical condensates and arbitrary bleach geometries, since spherical symmetry is only assumed for ease of data analysis. We were able to quantify the impact of neighboring droplets and the coverslip on the recovery dynamics. We found that neighboring droplets caused an appreciable speed-up in overall recovery, while emulating a coverslip caused a weak slow-down. In order to experimentally verify our theory, we have used three in vitro droplet systems, two composed of charged synthetic polymers and one with a purified protein component. There is remarkable quantitative agreement between our theory and the diffusion dynamics observed inside such droplets. This agreement shows that proteins and charged, synthetic polymers can form droplets that follow simple diffusive dynamics. Crucially, we use the full spatio-temporal data for fitting and can thus distinguish the timescale set by intra-droplet diffusion from the timescales at play in the dilute phase. We extract the intensity directly at the inside of the droplet interface and fit a spatially resolved diffusion equation to the ensuing recovery. We use the boundary intensity as a dynamic boundary condition and the experimentally measured profile as initial condition. Within the statistical fluctuations, the numerical solutions and the experimental data are not distinguishable (see *Figure 1b–d*). Throughout the time course, we find excellent agreement with the data and have thus found a method with minimal approximations that can precisely measure the inside diffusion coefficient $D_{in}$.

Building on the analysis inside the droplet, we show that there is a relationship between partition coefficient $P$ and the diffusion coefficient in the dilute phase, $D_{out}$. Data obtained from FRAP experiments define a line in ($D_{out}$, $P$) space, along which a range of parameter sets can reliably account for the boundary dynamics. This relationship allowed us to characterize changes in $P$ upon salt addition, opening an alternative avenue for characterizing $P$ without relying on fluorescence intensities. This is particularly important in light of recent results obtained by quantitative phase microscopy (QPM). These results show that measuring partition coefficients based on fluorescence intensity can lead to strong underestimation of $P$ (*McCall et al., 2020*). While it is tempting to interpret our values for $P$ as actual partition coefficients, we would like to stress that these values were obtained from a physical model, which does not consider additional effects at the condensate interface such as a potential interfacial resistance. This effect has recently been hypothesized to solve a contradiction in time scales between FRAP and FCS experiments (*Taylor et al., 2019*). However, introducing an interfacial resistance significantly lowered the fit quality in *Taylor et al., 2019*, an issue that has yet to be resolved. In this context, it will be interesting to reexamine work for three-component systems (*Binks and Lumsdon, 2000*; *Lin et al., 2003*). Some evidence for an interfacial resistance across a PEG/Dextran interface has been found for some types of molecules, for example DNA, bovine serum albumin, and bovine γ-globulin (*Hahn and Hardt, 2011*; *Gebhard et al., 2021*). Measuring $D_{out}$ via FCS, similar to *Taylor et al., 2019*, and partition coefficients via QPM (*McCall et al., 2020*), will allow further characterization of the existence of interfacial effects.

Our approach can be readily extended to multi-component systems with an arbitrary number of components, which is particularly useful in vivo. This would hardly be possible for techniques that do not use labeled components, such as QPM or other scattering methods. Of particular interest are multi-component systems with chemical reactions away from equilibrium. Our approach can then be

used to determine the diffusion coefficients and concentration levels of reactants, and thereby provide insights into reaction kinetics. Interestingly, introducing the bleached molecules via a ternary mixture also enabled us to derive the Langevin equation governing single-molecule motion in phase-separated media, thus providing a link to SPT (*Bo et al., 2021*). Approaches for single labeled molecules are relevant since high labeling fractions were shown to alter the viscosity and thus kinetics in dense protein phases (compare viscosity for PGL-3 in *Jawerth et al., 2018* with viscosity for tagged PGL-3:GFP *Jawerth et al., 2020* ). Finally, by accurately measuring $D_{\text{out}}$, our technique can also be employed to characterize rheological properties of condensates such as the recently reported glass-like dynamics of protein droplets (*Jawerth et al., 2020*).

## Materials and methods

### Coacervate assay
#### General reagents
Carboxymethyl-dextran sodium salt (CM-Dex, (C6H10O5)n.(COOH), 10–20 kDa, monomer MW = 191.3g/mol), Poly-L-lysine hydrobromide (PLys, (C6H12N2O)n, 4–15 kDa, monomer MW = 208.1g/mol) and adenosine 5'-triphosphate disodium salt hydrate (ATP, C10H14N5Na2O13P3, MW = 551.1g/mol) were purchased from Sigma Aldrich. FITC-PLys ((C6H12N2O)n.(C21H11NO5S), $25\,000\,\mathrm{g/mol}$) was purchased from Nanocs, NewYork, USA . Milli Q water was used to prepare aqueous stocks of CM-Dex (1000 mM, pH 8), PLys (200 mM, pH 8) and ATP (100 mM, pH 8). All solutions were stored in the freezer at -20 until use and the pH of all stocks was adjusted using a stock solution of 1M NaOH.

#### Coacervate preparation
Stock solutions of CM-Dex, PLys and ATP were first diluted to 25 mM and the PLys solution doped with 1% v/v PLys-FITC. Diluted solutions of CM-Dex/PLys or PLys/ATP were then mixed together at a 4:1 vol ratio (16 μl), resulting in the formation of turbid coacervate solutions. Solutions were left to equilibrate for at least 5 min before imaging, up to a maximum of 15 min when larger droplets were desired.

### PGL-3 droplets
PGL-3 was purified and stored as previously described (*Saha et al., 2016*). To obtain droplets, 300 mM KCl stock protein solution was diluted to the desired concentration, achieving final salt concentrations of 50–180 mM. A small imaging volume was created by using polystyrene beads, resulting in complete droplet sedimentation after less than five minutes. Droplets were imaged immediately to avoid changes in material properties due to ageing (*Jawerth et al., 2020*).

### Microscopy and FRAP
#### Confocal imaging
Droplets were imaged at midplane by visually defining the focal position with the largest droplet area of the droplet of interest. Images were acquired on an Andor spinning disk confocal microscope equipped with an Andor IX 81 inverted stand, a FRAPPA unit, an Andor iXON 897 EMCCD camera, and a 488 nm laser, using a 60x/1.2 U Plan SApo OLYMPUS water objective. Imaging conditions were optimized for minimal bleaching at the required frame rate. Frame rates were optimised for each system: PGL-3, $0.1\,\mathrm{s}<\Delta t<5\,\mathrm{s}$, CMD/PLYS, $\Delta t = 0.03\,\mathrm{s}$, PLYS/ATP, $\Delta t = 0.07\,\mathrm{s}$.

#### Frap
Droplets were bleached in their entirety by using the minimal FRAP ROI that encompasses the entire droplet. FRAP was performed in three focal planes, equally spaced across the droplet in z-direction, to reduce non-uniform bleaching of the droplet. FRAP rates and dwell times were chosen such that left-over fluorescence intensity above background was smaller than 1% for PGL-3 and smaller than 15% for coacervate droplets to maximize bleaching within the droplet while keeping bleaching impact on the droplet environment minimal.

## Data analysis

### Azimuthal averaging and normalization

Time-lapse images were cropped with the droplet of interest in the center. An azimuthal average was performed around the center of the droplet to obtain a 1D profile along the radial coordinate with minimal loss of data, using the *radialavg* function provided by David J. Fischer on Matlab File Exchange (*Fischer, 2016*). Camera background was subtracted uniformly from the resulting 1D profiles. The radial intensity profile at the prebleach stage was used for normalization and to correct for optical artefacts that lead to increased fluorescence at the droplet center compared to the droplet-bulk interface. Data close to the droplet interface cannot be used for fitting, since the droplet has an artificially broad boundary due to the point-spread function and likely due to curvature effects. Therefore, on average, the intensity of the ten pixels closest to the boundary were not used for analysis. The droplet boundary was defined as the inflection point of the azimuthally averaged profile in the pre-bleach frame.

Immediately after bleaching, a uniform recovery across the entire droplet can be seen, which cannot be spatially resolved even at frame rates $<30\,\text{ms}$. This recovery is fast compared to the recovery by diffusion from the outside for all systems under investigation. We thus chose to not account for this uniform recovery in our model and instead start the fitting after a time lag that depends on the system and droplet size. This offset typically consists of less than 5% of the total pre-bleach intensity. Additionally, bleaching is not complete, resulting in an additional offset above the camera background even immediately after bleaching (see gray lines in *Figure 1*).

Photo-bleaching due to continuous imaging was minimal in all droplet types. We thus chose to not account for imaging-induced photo-bleaching, in order to not introduce additional noise due to necessarily occurring fluctuations within the bleach correction.

### Extracting experimental boundary conditions

$c_u(r = R_-, t)$ was extracted from the intensity profiles as the value at the outermost pixel. In order to speed up fitting and avoid jumps in $c_u(r = R_-, t)$, the extracted intensity values were sorted to eliminate small fluctuations.

### Fitting of $D_{\text{in}}$ by using experimentally measured boundary conditions (*Figure 1*)

The resulting spatio-temporal profiles were fit as described in the main text, using $D_{\text{in}}$ as a single global fit parameter and using $c_u(r = R_-, t)$ as described above as the system's time-dependent boundary condition. Fits were performed in MATLAB (Mathworks), using pdepe to solve the PDE and fminsearch for minimizing the squared distance between data and model. Code is available at https://gitlab.pks.mpg.de/mesoscopic-physics-of-life/DropletFRAP (*Hubatsch, 2021a*; copy archived at swh:1:rev:7e5b59fff3c634cfce5d0f99a86c807635a090fd).

## Numerical solution of *Equation (6)*

*Equation 6* was solved using either pdepe (MATLAB (Mathworks), *Figures 4* and *5*, for spherically symmetric systems) or by using the finite element method via the FENICS environment (*Anders et al., 2012*) for arbitrary 3D geometries (*Figure 3*). All fits in *Figures 4* and *5* were performed using fminsearch based on a squared-difference metric. Code is available at https://gitlab.pks.mpg.de/mesoscopic-physics-of-life/frap_theory (*Hubatsch, 2021b*; copy archived at swh:1:rev:2c4a972a380df7f9e86ddbbf0ae921443ce0800f).

## Acknowledgements

We are grateful to P McCall, A Fritsch, JM Iglesias-Artola, G Bartolucci, T Wiegand, M Karnat, E Filippidi, T Harmon, F Jülicher and members of the Weber and Hyman groups for stimulating discussions, and P McCall and J Pfanzelter for very valuable and insightful comments on the manuscript. L Hubatsch, AA Hyman and C Weber acknowledge the SPP 2191 ''Molecular Mechanisms of Functional Phase Separation'' of the German Science Foundation for financial support.

## Additional information

### Funding

| Funder | Grant reference number | Author |
|---|---|---|
| Deutsche Forschungsgemeinschaft | SPP 2191 | Lars Hubatsch<br>Anthony A Hyman<br>Christoph A Weber |

The funders had no role in study design, data collection and interpretation, or the decision to submit the work for publication.

### Author contributions

Lars Hubatsch, Conceptualization, Data curation, Software, Formal analysis, Validation, Investigation, Visualization, Methodology, Writing - original draft, Writing - review and editing; Louise M Jawerth, Conceptualization, Resources, Validation, Writing - review and editing; Celina Love, Resources, Investigation; Jonathan Bauermann, Formal analysis, Validation, Methodology; TY Dora Tang, Resources, Funding acquisition, Writing - review and editing; Stefano Bo, Formal analysis, Supervision, Methodology; Anthony A Hyman, Conceptualization, Funding acquisition, Writing - review and editing; Christoph A Weber, Conceptualization, Supervision, Funding acquisition, Validation, Investigation, Methodology, Writing - original draft, Project administration, Writing - review and editing

### Author ORCIDs

Lars Hubatsch  https://orcid.org/0000-0003-1934-7437
Louise M Jawerth  https://orcid.org/0000-0002-7221-939X
Jonathan Bauermann  http://orcid.org/0000-0002-0301-7655
Stefano Bo  https://orcid.org/0000-0002-2738-867X
Christoph A Weber  https://orcid.org/0000-0001-6279-0405

### Decision letter and Author response

Decision letter https://doi.org/10.7554/eLife.68620.sa1
Author response https://doi.org/10.7554/eLife.68620.sa2

## Additional files

### Supplementary files

• Source data 1. Quantification_Fig1_Fig4.

• Transparent reporting form

### Data availability

Code for modelling and data analysis is available at https://gitlab.pks.mpg.de/mesoscopic-physics-of-life/frap_theory and https://gitlab.pks.mpg.de/mesoscopic-physics-of-life/DropletFRAP (copies archived at https://archive.softwareheritage.org/swh:1:rev:2c4a972a380df7f9e86ddbbf0ae921443-ce0800f and https://archive.softwareheritage.org/swh:1:rev:7e5b59fff3c634cfce5d0f99a86-c807635a090fd, respectively).

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

## Appendix 1

### Limit of narrow interfaces

Here, we derive the effective droplet model for our dynamic equation

$$\partial_t c_u = \nabla \cdot \left[ D(\phi_{\text{tot}}) \left( \nabla c_u - c_u \frac{\nabla \phi_{\text{tot}}}{\phi_{\text{tot}}} \right) \right] \tag{11}$$

$$= -\nabla \cdot \boldsymbol{j}_{\text{u}} \tag{12}$$

by considering the limit of narrow interfaces. In the equation above, $\boldsymbol{j}_{\text{u}}$ denotes the flux of unbleached molecules. Conservation of molecules at the interface implies

$$\lim_{\ell \to \epsilon} \boldsymbol{j}_{\text{u}}(R - \ell) = \lim_{\ell \to \epsilon} \boldsymbol{j}_{\text{u}}(R + \ell) \,, \tag{13}$$

where $\ell$ is the characteristic size of the interface. Moreover, $\epsilon > 0$ is a small but non-zero parameter by which we define the position directly left and right of the interface. In this limit, $\nabla \phi_{\text{tot}}|_{R \pm \epsilon} = 0$ directly left and right of the interface. Thus, we obtain *Equation (8c)* for narrow interfaces. However, the contribution $D(\phi_{\text{tot}}) c_u \nabla \phi_{\text{tot}} / \phi_{\text{tot}}$ in *Equation (11)* that determines the dynamics through the interface implies a Dirichlet boundary condition for the concentration of unbleached molecules at the interface $r = R$. Since $\nabla \phi_{\text{tot}} / \phi_{\text{tot}} \propto \ell$, we demand

$$\lim_{\ell \to 0} \left( \ell c_u|_{R - \ell} \frac{\nabla \phi_{\text{tot}}}{\phi_{\text{tot}}}|_{R - \ell} \right) = \lim_{\ell \to 0} \left( \ell c_u|_{R + \ell} \frac{\nabla \phi_{\text{tot}}}{\phi_{\text{tot}}}|_{R + \ell} \right) \tag{14}$$

in the limit of decreasing interface width $\ell$. This condition ensures that the dynamics at the interface remain unchanged for decreasing $\ell$. Parameterizing the interface for example by $\phi_{\text{tot}}(r) = \phi_{\text{out}}^{\text{eq}}[1 + (P - 1)(1 + \tanh(r/\ell))/2]$, we find that *Equation (14)* leads to *Equation (8d)*.

## Appendix 2

### Solution of effective droplet model for FRAP

In this appendix, we derive the solution for the two diffusion equations coupled at the interface for a spherically symmetric case; see *Equations (8)*. We consider an initial condition with a constant concentration $c(r,0) = c_0$ outside a droplet with radius $R$ and bleaching gives rise to $c(r,0) = 0$ inside the droplet. Performing a Laplace transformation, $\hat{c}(r,s) = \int_0^\infty dt\, e^{-st} c(r,t)$, of *Equations (8)*, we find

$$s\hat{c}_u(r,s) = D_{\text{in}}\nabla^2 \hat{c}_u(r,s), \qquad \text{for } r < R, \tag{15a}$$

$$s\hat{c}_u(r,s) = D_{\text{out}}\nabla^2 \hat{c}_u(r,s) + c_0, \quad \text{for } r \geq R. \tag{15b}$$

where $s$ is the rate parameter of the Laplace transform. The corresponding solutions read

$$\hat{c}_u(r,s) = \begin{cases} \frac{a}{r}\sinh(\xi_{\text{in}}r), & \text{for } r < R, \\ \frac{b}{r}\exp(-\xi_{\text{out}}r) + \frac{c_0}{s}, & \text{for } r \geq R, \end{cases} \tag{16}$$

where $\xi_{\text{in/out}} = \sqrt{s/D_{\text{in/out}}}$. We have selected the solutions with no radial flux at the origin and a finite concentration at infinity. The remaining unknown constants $a$ and $b$ are determined by the conditions at the interface stated in *Equations (8)*. Inside the droplet $r < R$, the solution reads

$$\hat{c}_u(r,s) = \frac{c_0 PR(D_{\text{out}} + R\sqrt{D_{\text{out}}s})}{rs\left(D_{\text{out}} - PD_{\text{in}} + R\sqrt{D_{\text{out}}s} + PR\sqrt{D_{\text{in}}s}\coth\left(R\sqrt{\frac{s}{D_{\text{in}}}}\right)\right)} \frac{\sinh\left(r\sqrt{\frac{s}{D_{\text{in}}}}\right)}{\sinh\left(R\sqrt{\frac{s}{D_{\text{in}}}}\right)}. \tag{17}$$

Even though it is non-trivial to perform the inverse Laplace transformation of this expression we can obtain some analytical understanding of the behavior at short and long timescales by considering the asymptotics of large and small $s$. Expanding *Equation (17)* for small $s$, we obtain:

$$\hat{c}_u(r,s) = c_0 P\left[\frac{1}{s} - \frac{\left(2P\frac{D_{\text{in}}}{D_{\text{out}}} + 1\right)R^2 - r^2}{6D_{\text{in}}}\right] + O(s^{1/2}). \tag{18}$$

This relationship shows that for small $s$, the rescaled concentration $\hat{c}_u(r,s)/P$ is affected by $D_{\text{out}}$ only via the $PD_{\text{in}}/D_{\text{out}}$ combination. Conversely, for large $s$, the leading contribution features the term

$$\hat{c}_u(r,s) \sim \frac{c_0 PR}{rs}\left(1 + P\sqrt{\frac{D_{\text{in}}}{D_{\text{out}}}}\right)^{-1} e^{(r-R)\sqrt{\frac{s}{D_{\text{in}}}}} \tag{19}$$

showing that for short times the system is influenced by $D_{\text{out}}$ via the ratio $P\sqrt{D_{\text{in}}/D_{\text{out}}}$. Note that for these short times the solution is well approximated by that of a one-dimensional system. These long and short time behaviors are confirmed by the numerical solutions plotted in *Appendix 2—figure 1*. C, which show that the evolution of the rescaled concentrations match at short times, if $P\sqrt{D_{\text{in}}/D_{\text{out}}}$ is kept constant but start to deviate for longer times. Conversely, at longer times the dynamics are invariant if $PD_{\text{in}}/D_{\text{out}}$ is kept constant except for a shift due to a short timescale transient.

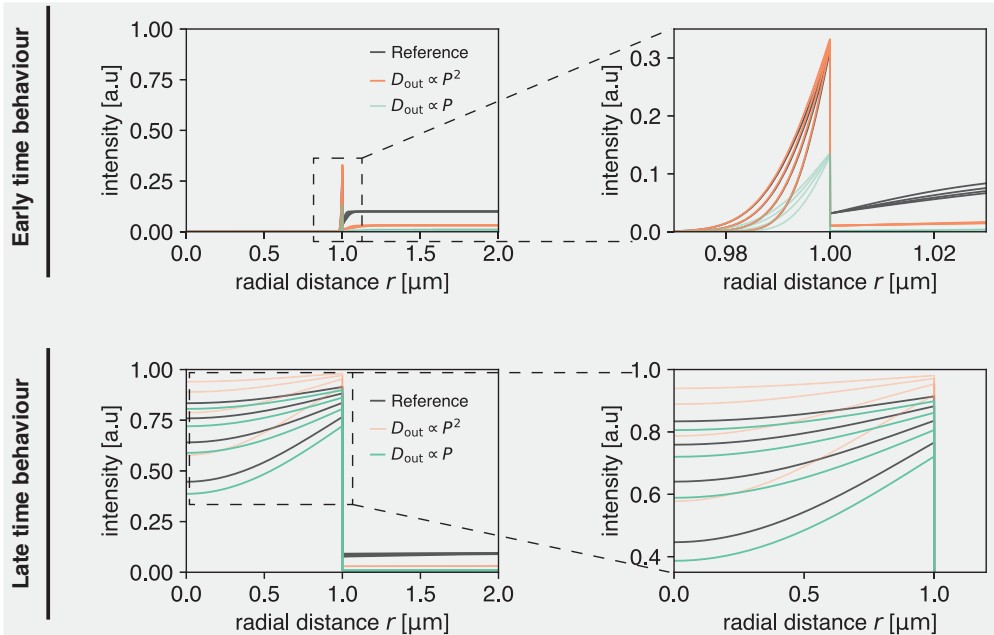

**Appendix 2—figure 1.** Solution to dynamics inside droplets is unique, but early and late time behavior can be approximated by different scaling regimes. (Top) Early time behavior of numerical solutions. If a parameter set is chosen ('Reference'), at early times this can be approximately recovered by scaling $D_{out} \propto P^2$ (orange). A linear scaling does not recover the same inside dynamics at early times (green). (Bottom) Late time behavior of numerical solutions. At late times, the reference dynamics can be approximately recovered by scaling $D_{out} \propto P$ (green). Meanwhile, $D_{out} \propto P^2$ does not recover the reference inside dynamics (orange).

