## [Decision Letter]

**Acceptance summary:**

This work is timely and relevant as the field grapples with the issue of diffusive dynamics across phase boundaries. The numerical formalism in this work will be of broad interest to the condensate field.

**Decision letter after peer review:**

Thank you for submitting your article "Quantitative Theory for the Diffusive Dynamics of Liquid Condensates" for consideration by *eLife*. Your article has been reviewed by 2 peer reviewers, including Rohit Pappu as the Reviewing Editor and Reviewer #1, and the evaluation has been overseen by José Faraldo-Gómez as the Senior Editor.

Both reviewers have arrived at similar conclusions. FRAP is a method that is routinely used to study internal dynamics of molecules within biomolecular condensates. Of course, FRAP was introduced in a completely different context, and as has been shown before, its adoption needs appropriate adaptation to the context of interest. How should FRAP data be analyzed in the context of studying biomolecular condensates that form via phase separation. The authors build on the work of Hardt and coworkers, and demonstrate the incorporation of a Flory-Huggins free energy alongside diffusion equations to describe the dynamics of unbleached molecules, using features of mass balance. They show that numerical solutions of the derived equations – see Equation (6) – can be used to fit FRAP data for different systems. The authors also suggest that inferences from FRAP data can go beyond extraction of individual parameters. In other words, FRAP data seem to be more information-rich than originally thought. The current version, although very interesting, features opacities that should be remediable by following the recommendations made by both reviewers.

Essential revisions:

1) Two distinct flowcharts summarizing how Equations (1) and (6) are used in the fitting of FRAP data will be essential.

2) Accounting for interfacial tension and / or interfacial resistance (see Taylor et al.) requires discussion.

3) Please delete references to non-equilibrium situations since the model imposes detailed balance throughout. In fact, this point should be made clear.

4) There is considerable confusion regarding the claims regarding being able to extract D_in_, D_out_, and P from single sets of FRAP data and the actual demonstration of this versatility. This is accentuated by considerable confusion caused, for both reviewers, by the introduction of the cost function, which was opaque, and the sweep of parameters for D_out_ and P that clearly give satisfactory fits to the FRAP data. At this juncture, the claim of being able to extract more insights from FRAP data than one is accustomed to seeing has not been unequivocally demonstrated.

5) Several scholarly issues, specifically pertaining to the work of Hardt and colleagues, and the semantics of what constitutes a phenomenological vs. physical theory description were raised. These should be addressed and there is a strong desire to see a toning down of what were perceived as over-claims.

*Reviewer #1 (Recommendations for the authors):*

As it currently stands, the average practitioner of FRAP is likely to find the narrative to be rather opaque. Two flowcharts, that summarize (a) how the dynamic boundary condition and its application lead to the internal diffusion coefficients and (b) the use of equation (6) for experimentalists in their analysis of FRAP data need to be added. In doing so, it is really important to explain when and how the analysis can be used, and when and where it cannot be used.

Other recommendations:

1. phi is a conserved order parameter, because the underlying theory imposes a closed system. Therefore, the relations between phi_tot_ and phi_u_ etc. come not from incompressibility, but from mass balance in a closed system.

2. The work of Steffen Hardt, mentioned in passing, and labeled as being phenomenological needs rectification. Elaboration of their work and a clear, scholarly contrast between the current effort, the published work of Hardt that goes beyond the 2008 Langmuir paper, and the differences between the work of Taylor et al., would be helpful for the reader who is interested in understanding what is new, and what is different. In this context, please also see Lin et al., (2003) Science 299: 226, Binks and Lumsdon (2000) Langmuir 16: 8622.

3. The authors classify previous efforts as being phenomenological and the current effort as a physical theory. Respectfully, I disagree with this characterization. Inasmuch as previous efforts largely use the same equations as being used by the authors, with the key difference being the use of the FH functional for extracting the chemical potentials, the extant literature and the current manuscript are both phenomenological or both physical theories. It would be best to drop these prefaces and simply state what has been done. Specifically, the work of Hardt and coworkers does account for phase boundaries. They don't use the FH free energy functional, and this is really the only main difference. And many would rightly argue that the FH functional is phenomenological. So, please do not use terms like "first principles" for this work, and phenomenological to inadvertently dismiss other efforts.

4. Throughout the manuscript, there is a lot of self-congratulation about the excellent fits, the striking agreements etc. This is jarring. It is impossible to know if one should be impressed with the agreements or if such agreements are readily achieved. An unvarnished statement of the facts is sufficient.

5. The description of the cost function analysis needs a lot of work. In effect, the implication is that D_out_ depends on P. Whether this is an implicit function or not is unclear. The cost function has not been defined in the main text, and why there should be a minimum is unclear. There is also a curious leap that happens: at one point, we see that D_out_ and P that describe the data can span a wide range. This would lead to inference that the FRAP data cannot pin down these parameters absent independent measurements. However, the conclusions paint a more rosy picture, suggesting that knowledge of D_in_ is sufficient, and that fitting the FRAP data will lead to reliable estimates of D_out_ and P. The section that describes this leap needs a lot of work. The flowcharts recommended in the public review will help.

6) There are important caveats that apply as well. The slow or non-recovery of fluorescence, becomes a scenario wherein D_in_ will be very small or immeasurable. Alternatively, if the material state has changed, then the usage of the dynamic boundary condition might still yield reliable measurements of D_in_, due to an anisotropic stress tensor, but this would be an erroneous estimate of D_in_ on the inside. Therefore, it is important to lay out a set of requirements for the applicability of the proposed approach, i.e., under what circumstances can the method be used, and under what circumstances can it not be used.

7) As a logical follow up to the preceding point, how might one falsify the theory? Fitting FRAP data is inadequate for such an effort. This will be very useful for experimentalists and computational scientists alike.

8) Finally, do the authors know that the salt concentrations across the phase boundary are equal to one another or is this implicitly assumed? This is particularly relevant for the synthetic coacervates. Sing and coworkers have proposed that there are differences in salt concentrations (even ion concentrations) across phase boundaries. This is relevant because it requires the addition of a Donnan potential to the equations that describe molecular transport. A discussion of this issue would be helpful.

*Reviewer #2 (Recommendations for the authors):*

Concerns on figures and writing:

1. Recurrent lack of clarity and/or consistency in how figures are presented

When dissecting the data shown in figures, we often found details in presentation that detracted from our understanding of the study. Notable, but not exhaustive, examples are listed:

a. The retrieval of physical parameters in this study frequently alternates between the use of Equation 1 to track FRAP data, and Equation 6 to model droplet dynamics. The authors do not clearly distinguish the purpose of these equations in the theoretical sections, leaving the reader to understand by context.

b. The data in Figure 3 would benefit from control data sets that entirely lack the coverslip and/or neighbouring droplets for comparison. In particular, showing the retrieval of identical Din across all simulations (including no coverslip nor neighboring droplets condition) will convey the major point of the paper very clearly.

c. In the caption of Figure 5 it is unclear why the authors have denoted the four indicated values of Dout and P as "reference systems' as opposed to example parameter sets.

We recommend the authors reassess their figures for clarity of the information that is meant to be communicated.

2. The theory sections lack needed elaboration in some areas.

a. The authors conclude starting on line 153 that P, Din, and Dout can be treated as independent of each other for sufficiently large P. Their justification is that the unknown mobility functions do not impose constraint on Dout(P) other than shown in Equation 10. While the data demonstrates that this is a valid assumption, we find this justification to be opaque and would like to see further elaboration on how independent P, Din, and Dout follows from the mobility functions being unknown.

Suggestions to the authors for bolstering the overall strength of this study below:

1. When interpreting the Results section, we often found that the nature of the experiment was unclear. This is particularly true for Figure 4 and 5. We recommend specifying the procedure followed to acquire the data more clearly and explicitly.

2. In Figure 1c, the label "Dynamic BC" made this figure confusing to interpret, especially because there is a same-color arrow depicting time progression. The authors should consider some other way of noting that dynamic BC is applied at the max r (=R-). Also, for the line going across the earliest data points, "initial condition" rather than the "fit" label may be more appropriate.

3. If available, showing corrected viscosity data from Jawerth et al., 2020 rather than Jawerth et al., 2018 in Figure 1d would bolster this figure via internal consistency with the text in line 98. We would like to be able to back-calculate consistent viscosities using your diffusion data and the Stokes-Sutherland-Einstein relationship.

4. Since the time progression of FRAP recovery is illustrated in Figure 2c, we believe the time point label in Figure 2 to be unnecessary and possibly confusing.

5. We find that the citation of privately communicated and unseen data in line 234 does not add to the preceding statement. Optimum salt concentration is a very believable observation.

6. There appears to be a typo in the subscripts on line 138. Both read "in".

7. Regarding the concluding statement on line 306: We do not find that Jawerth et al., 2018,2020 contain discussions of altering dense phase kinetics by high labeling fraction. Rather, McCall et al., 2020 describes effects closer to this.

8. Regarding Figure 5a, labelling the "ratio" of Dout and P can be confusing. We suggest noting the specific (Dout, P) pairs to specify the points. Also, a clear distinction between "simulation generating parameters (points)" and the "Dout and P dependence (lines) obtained from simulation result and equation 6" is needed.

9. Please consider labelling the point in Figure 5b as "simulation input" or "simulation parameter" rather than "reference simulation".

10. We suggest the following change in line 230 for clarity: "Specifically, for salt concentrations in the range from 50 mM to 180 mM, we find that the estimated partition coefficient P of PGL-3 droplets decreases more than 10-fold."

11. The significance of the shading in Figure 4d is unclear and inconsistent with how the analogous data in Figure 4e is presented.

12. The text accompanying Figure 5 beginning on line 249 describes the range of Dout and P used as "relevant for protein condensates and coacervate droplets' without citation. We recommend backing up the validity of the range of Dout and P used here with evidence from literature.

13. In Figure 1b, individual FRAP recovery curves cut off at seemingly arbitrary points and are difficult to distinguish based on provided color coding. We also note that the curve of 100mM salt concentration does not seem to fit the same trend as the other data sets, but is not discussed by the authors.

14. Figure 1f shows the precision of Din determination for the two coacervate systems, but not the protein condensate system, and a salt concentration for comparison with Figure 1d is not provided.

---

## [Author Response]

Essential revisions:1) Two distinct flowcharts summarizing how Equations (1) and (6) are used in the fitting of FRAP data will be essential.

We are thankful for this suggestion and have added the proposed flow charts; see Figure 1 and Figure 4 in revised manuscript.

2) Accounting for interfacial tension and / or interfacial resistance (see Taylor et al.) requires discussion.

We now discuss our results in the light of a potential interfacial resistance at the phase boundary. We have also now made explicitly clear in several key parts of the paper that we assume no interfacial resistance. For a more detailed discussion, see also replies below. Interfacial tension is related to the interfacial width which is accounted for in our model; see \ell in the paragraph below Equation (6).

3) Please delete references to non-equilibrium situations since the model imposes detailed balance throughout. In fact, this point should be made clear.

We agree that in our manuscript we focus on the case where the total volume fraction (composed of bleached and unbleached molecules) is at thermodynamic equilibrium leading to partial_t_ phi_tot_ = 0. Please note that our theory can be applied to non-equilibrium situations, i.e., in the presence of fluxes. For example, see e.g. Equation (6) in Bo et al., (reference in manuscript), where we use this coarse-grained theory to derive a single molecule description also away from thermodynamic equilibrium.

Based on the reviewer’s remark, we have revised the paragraph around Equation (2) and now stress more clearly that we focus on the cases where detailed balance holds.

4) There is considerable confusion regarding the claims regarding being able to extract D_in_, D_out_, and P from single sets of FRAP data and the actual demonstration of this versatility. This is accentuated by considerable confusion caused, for both reviewers, by the introduction of the cost function, which was opaque, and the sweep of parameters for D_out_ and P that clearly give satisfactory fits to the FRAP data. At this juncture, the claim of being able to extract more insights from FRAP data than one is accustomed to seeing has not been unequivocally demonstrated.

We have now made more explicit that this refers to a theoretical possibility of extracting all three parameters, by introducing a new section heading and adding descriptive text. Also, the flow charts should help, in particular regarding the cost function. We have reevaluated our conclusion regarding being able to extract P if D_out_ is known and vice versa. While we still think this is a promising avenue, we agree that the current data are better interpreted more carefully. We now conclude that we find a unique relationship between P and D_out_ for every condition, which allows us to extract relative changes in P for different salt conditions. This was not possible previously.

5) Several scholarly issues, specifically pertaining to the work of Hardt and colleagues, and the semantics of what constitutes a phenomenological vs. physical theory description were raised. These should be addressed and there is a strong desire to see a toning down of what were perceived as over-claims.

We have carefully revised and extended the discussion of the previous literature, such as the works of the Hardt group and Taylor et al. Moreover, we have removed statements such as “phenomenological” and similar formulations.

Reviewer #1 (Recommendations for the authors):As it currently stands, the average practitioner of FRAP is likely to find the narrative to be rather opaque. Two flowcharts, that summarize (a) how the dynamic boundary condition and its application lead to the internal diffusion coefficients and (b) the use of equation (6) for experimentalists in their analysis of FRAP data need to be added. In doing so, it is really important to explain when and how the analysis can be used, and when and where it cannot be used.

We have added the flow charts according to the reviewer’s suggestion. Moreover, we say explicitly when our analysis can be used and when not (see text below Equation 1).

Other recommendations:1. phi is a conserved order parameter, because the underlying theory imposes a closed system. Therefore, the relations between phi_tot_ and phi_u_ etc. come not from incompressibility, but from mass balance in a closed system.

Our system conserves particle number of each component after photobleaching. Since we consider an incompressible system, the volume fractions of bleached and unbleached molecules add up to the total volume fraction at each position (Equation (2) in manuscript). Please note that the latter is also true for open systems that are incompressible.

2. The work of Steffen Hardt, mentioned in passing, and labeled as being phenomenological needs rectification. Elaboration of their work and a clear, scholarly contrast between the current effort, the published work of Hardt that goes beyond the 2008 Langmuir paper, and the differences between the work of Taylor et al., would be helpful for the reader who is interested in understanding what is new, and what is different. In this context, please also see Lin et al., (2003) Science 299: 226, Binks and Lumsdon (2000) Langmuir 16: 8622.

We are grateful for the additional references and now discuss the work of Hardt and colleagues more thoroughly, covering additional references. Note, both the works by Hardt and colleagues as well as Lin/Science and Binks/Langmuir, assume a third species that adsorbs to the interface. However, we consider a binary mixture of protein and solvent and currently have no evidence that the protein component exists in several states. While we also find the possibility of an interfacial resistance compelling, we didn’t find any direct evidence of it.

We cite Taylor et al., on several occasions in the manuscript, where we felt it was appropriate. They provide a very nice overview of currently used heuristic fit functions and their (severe) limitations. They also give an effective droplet model (referenced), without derivation. However, as discussed in their work, their fitting yields unrealistic results, and even after introducing an additional fit parameter (interfacial resistance), the fits get worse, instead of better. This is curious, given that the interfacial tension model is a superclass of the model without interfacial tension and should thus fit at least as well, if not better than the original model. In our method to determine D_in_ (Figure 1 and Equation (1)), we completely circumvent any boundary effects, can measure D_in_ reliably by fitting the spatial profiles, and thus provide a significant advance. Figures 3-5 have no overlap with any of the references.

3. The authors classify previous efforts as being phenomenological and the current effort as a physical theory. Respectfully, I disagree with this characterization. Inasmuch as previous efforts largely use the same equations as being used by the authors, with the key difference being the use of the FH functional for extracting the chemical potentials, the extant literature and the current manuscript are both phenomenological or both physical theories. It would be best to drop these prefaces and simply state what has been done. Specifically, the work of Hardt and coworkers does account for phase boundaries. They don't use the FH free energy functional, and this is really the only main difference. And many would rightly argue that the FH functional is phenomenological. So, please do not use terms like "first principles" for this work, and phenomenological to inadvertently dismiss other efforts.

We have now dropped all occurrences of ‘phenomenological’ when referring to the work of others.

4. Throughout the manuscript, there is a lot of self-congratulation about the excellent fits, the striking agreements etc. This is jarring. It is impossible to know if one should be impressed with the agreements or if such agreements are readily achieved. An unvarnished statement of the facts is sufficient.

We have toned down our language. The fits however, are “excellent” in our eyes, since we get almost complete agreement with the data within the statistical fluctuations due to radial averaging. From the fits in Figures 1 b-d, it is clear that there are barely any systematic deviations between model and data. Almost all deviations stem from statistical fluctuations of neighbouring pixels. These get stronger towards r=0, since fewer pixels can be used for radial averaging. In this context, please also note that even integrated intensity curves with more parameters are less well described by existing models (cf. Taylor et al.,).

5. The description of the cost function analysis needs a lot of work. In effect, the implication is that D_out_ depends on P. Whether this is an implicit function or not is unclear. The cost function has not been defined in the main text, and why there should be a minimum is unclear. There is also a curious leap that happens: at one point, we see that D_out_ and P that describe the data can span a wide range. This would lead to inference that the FRAP data cannot pin down these parameters absent independent measurements. However, the conclusions paint a more rosy picture, suggesting that knowledge of D_in_ is sufficient, and that fitting the FRAP data will lead to reliable estimates of D_out_ and P. The section that describes this leap needs a lot of work. The flowcharts recommended in the public review will help.

We have now added the requested flow charts and believe they will be very helpful. The distinction between what is currently feasible based on experiments (namely only getting an approximate relationship between D_out_ and P) and what is theoretically possible with additional assumptions and perfect data has now been made clearer by introducing an extra section heading and additional descriptive text.

6) There are important caveats that apply as well. The slow or non-recovery of fluorescence, becomes a scenario wherein D_in_ will be very small or immeasurable. Alternatively, if the material state has changed, then the usage of the dynamic boundary condition might still yield reliable measurements of D_in_, due to an anisotropic stress tensor, but this would be an erroneous estimate of D_in_ on the inside. Therefore, it is important to lay out a set of requirements for the applicability of the proposed approach, i.e., under what circumstances can the method be used, and under what circumstances can it not be used.7) As a logical follow up to the preceding point, how might one falsify the theory? Fitting FRAP data is inadequate for such an effort. This will be very useful for experimentalists and computational scientists alike.

We agree that our method has restricted applicability. As indicated by the reviewer, the underlying kinetic equation does no more apply when the condensed phase is for example a complex liquid with more than one time-scale on the time-scale of the recovery kinetics. However, we think that in this case shapes of spatial profiles should differ between such a complex liquid condensate and a simple liquid. We have added a paragraph to the Results section, pointing out the restricted applicability (below Equation (1)).

8) Finally, do the authors know that the salt concentrations across the phase boundary are equal to one another or is this implicitly assumed? This is particularly relevant for the synthetic coacervates. Sing and coworkers have proposed that there are differences in salt concentrations (even ion concentrations) across phase boundaries. This is relevant because it requires the addition of a Donnan potential to the equations that describe molecular transport. A discussion of this issue would be helpful.

The referee raises an interesting point. Indeed, theoretical studies suggest that salt concentrations can be different, though weakly, inside and outside of condensates (as shown for example in Sing 2020 JCP). Due to the large differences in molecular/particle volumes between salt ions and coacervate components / proteins, however, only weak differences of salt concentrations between phases are expected. Thus, we feel confident that the assumption of equal salt concentrations in both phases represents a reasonable approximation, at least for our systems. In addition, our results are consistent with unpublished data by Fritsch and colleagues.

Reviewer #2 (Recommendations for the authors):Concerns on figures and writing:1. Recurrent lack of clarity and/or consistency in how figures are presentedWhen dissecting the data shown in figures, we often found details in presentation that detracted from our understanding of the study. Notable, but not exhaustive, examples are listed:a. The retrieval of physical parameters in this study frequently alternates between the use of Equation 1 to track FRAP data, and Equation 6 to model droplet dynamics. The authors do not clearly distinguish the purpose of these equations in the theoretical sections, leaving the reader to understand by context.

We think that this should now be clear from the flow charts and figure legends.

b. The data in Figure 3 would benefit from control data sets that entirely lack the coverslip and/or neighbouring droplets for comparison. In particular, showing the retrieval of identical Din across all simulations (including no coverslip nor neighboring droplets condition) will convey the major point of the paper very clearly.

Panels c and f serve this purpose. In fact, box size is 8*8*8 um, such that the case of h=4um (panel b) gives almost the same results as the case of no coverslip and no neighbors. However, even in the scenario with largest influence of coverslip and neighbours (droplet very close to coverslip, neighbours very close to droplet) the fit can barely be distinguished from the simulated data set (c and f). This is now made clear in the legend.

c. In the caption of Figure 5 it is unclear why the authors have denoted the four indicated values of Dout and P as "reference systems' as opposed to example parameter sets.

This has been addressed now, please see below.

We recommend the authors reassess their figures for clarity of the information that is meant to be communicated.2. The theory sections lack needed elaboration in some areas.a. The authors conclude starting on line 153 that P, Din, and Dout can be treated as independent of each other for sufficiently large P. Their justification is that the unknown mobility functions do not impose constraint on Dout(P) other than shown in Equation 10. While the data demonstrates that this is a valid assumption, we find this justification to be opaque and would like to see further elaboration on how independent P, Din, and Dout follows from the mobility functions being unknown.

We are sorry for not being clear on this. As stressed in the manuscript and as shown for example by Equation (10), P, Din and Dout are not independent. If we knew the mobility as a function of phi_tot_, we could use e.g. Equation (10) to calculate from two of the three parameters the third one. However, mobility as a function of phi_tot_ is not known. We have now revised the paragraphs around Equation (10).

Suggestions to the authors for bolstering the overall strength of this study below:1. When interpreting the Results section, we often found that the nature of the experiment was unclear. This is particularly true for Figure 4 and 5. We recommend specifying the procedure followed to acquire the data more clearly and explicitly.

We have added a flow chart describing the procedure in figure 4 and hope that the experimental procedure is now clearer.

2. In Figure 1c, the label "Dynamic BC" made this figure confusing to interpret, especially because there is a same-color arrow depicting time progression. The authors should consider some other way of noting that dynamic BC is applied at the max r (=R-). Also, for the line going across the earliest data points, "initial condition" rather than the "fit" label may be more appropriate.

Following the reviewer’s suggestions, Figure 1 has been redone, including the addition of a flowchart.

3. If available, showing corrected viscosity data from Jawerth et al., 2020 rather than Jawerth et al., 2018 in Figure 1d would bolster this figure via internal consistency with the text in line 98. We would like to be able to back-calculate consistent viscosities using your diffusion data and the Stokes-Sutherland-Einstein relationship.

Unfortunately, Jawerth et al., 2020 only investigate c_salt_=75mM, we can thus not correct the entire graph, but can only note an order of magnitude given the available data point.

4. Since the time progression of FRAP recovery is illustrated in Figure 2c, we believe the time point label in Figure 2 to be unnecessary and possibly confusing.

We have shifted this information to the legend and removed the label.

5. We find that the citation of privately communicated and unseen data in line 234 does not add to the preceding statement. Optimum salt concentration is a very believable observation.

We have deleted the superfluous statement.

6. There appears to be a typo in the subscripts on line 138. Both read "in".

Corrected.

7. Regarding the concluding statement on line 306: We do not find that Jawerth et al., 2018,2020 contain discussions of altering dense phase kinetics by high labeling fraction. Rather, McCall et al., 2020 describes effects closer to this.

Here, we are referring to the change in droplet viscosity, when labelling PGL-3 with GFP vs unlabelled PGL-3 (Figure S5G in Jawerth et al., 2020 vs Figure 3G in Jawerth et al., 2018). This has been clarified.

8. Regarding Figure 5a, labelling the "ratio" of Dout and P can be confusing. We suggest noting the specific (Dout, P) pairs to specify the points. Also, a clear distinction between "simulation generating parameters (points)" and the "Dout and P dependence (lines) obtained from simulation result and equation 6" is needed.

We have deleted the legend and noted the parameter values in the figure caption. A sentence was added to the caption to distinguish open circles and solid lines.

9. Please consider labelling the point in Figure 5b as "simulation input" or "simulation parameter" rather than "reference simulation".

We went for ‘Example system’, trying to stay coherent with panel a.

10. We suggest the following change in line 230 for clarity: "Specifically, for salt concentrations in the range from 50 mM to 180 mM, we find that the estimated partition coefficient P of PGL-3 droplets decreases more than 10-fold."

This has been incorporated.

11. The significance of the shading in Figure 4d is unclear and inconsistent with how the analogous data in Figure 4e is presented.

We have removed the shading in 4d.

12. The text accompanying Figure 5 beginning on line 249 describes the range of Dout and P used as "relevant for protein condensates and coacervate droplets' without citation. We recommend backing up the validity of the range of Dout and P used here with evidence from literature.

We have added two references for P. D_out_ can be seen for example in Figure 1.

13. In Figure 1b, individual FRAP recovery curves cut off at seemingly arbitrary points and are difficult to distinguish based on provided color coding. We also note that the curve of 100mM salt concentration does not seem to fit the same trend as the other data sets, but is not discussed by the authors.

Figure 1b was previously shown to illustrate the dynamic boundary condition and to indicate a trend in recovery speed vs. salt concentration, which mostly holds, except for the noted outlier. However, to be able to quantitatively compare droplet recovery speed based on integrated fluorescence, droplets need to be of the same size, which is not entirely fulfilled in this graph. Since it seems to also have caused confusion with the overall fitting procedure, we chose to remove this panel and instead opt for a cleaner presentation of a single curve in Figure 1a.

14. Figure 1f shows the precision of Din determination for the two coacervate systems, but not the protein condensate system, and a salt concentration for comparison with Figure 1d is not provided.

Since we did not have a large enough sample size for any individual condition for PGL-3, we initially chose to not show a graph that corresponds to Figure 1f (now 1g). Here in Author response image 1 we include the requested graph for the reviewer’s convenience. We agree that it is useful to give a measure of measurement precision also for the protein system and now discuss this in the text. The average CoV for PGL-3 across all measurements is approximately four times larger than for the coacervates, likely due to variations in the experimental assay (see text).
